# Supporting islet function in a PVDF membrane based macroencapsulation delivery device by solvent non-solvent casting using PVP

**Denise F. A. de Bont**[1], **Sami G. Mohammed**[1], **Rick H. W. de Vries**[1], **Omar Paulino da Silva Filho**[1], **Vijayaganapathy Vaithilingam**[1], **Marlon J. Jetten**[1], **Marten A. Engelse**[2], **Eelco J. P. de Koning**[2,3], **Aart A. van Apeldoorn**[1]*

**1** Cell Biology-Inspired Tissue Engineering (cBITE), MERLN Institute for Technology-Inspired Regenerative Medicine, Maastricht University, Maastricht, The Netherlands, **2** Department of Nephrology, Leiden University Medical Center, Leiden, The Netherlands, **3** Hubrecht Institute, Utrecht, The Netherlands

* a.vanapeldoorn@maastrichtuniversity.nl

## Abstract

Type 1 diabetic (T1D) patients are life-long dependent on insulin therapy to keep their blood glucose levels under control. An alternative cell-based therapy for exogenous insulin injections is clinical islet transplantation (CIT). Currently the widespread application of CIT is limited, due to risks associated with the life-long use of immunosuppressive drugs to prevent rejection of donor cells. An immunoprotective macroencapsulation device can protect allogeneic islet cells against the host immune system and allow exploring extrahepatic transplantation sites. We report on the characterization and creation of porous polyvinylidene fluoride (PVDF) membrane-based devices intended for islet and beta-cell transplantation. We hypothesize that by incorporating polyvinyl-pyrrolidone (PVP) into a PVDF solution the permeability of PVDF membranes for insulin and glucose can be improved by solvent-non solvent casting to create submicrometer porous films. We show that the use of water-soluble PVP, can significantly increase glucose diffusion through these membranes while still having the ability to block immune cells from migrating through these membranes. Human donor islets loaded into devices made from these thin PVDF/PVP membranes showed a 92 ± 4% viability after 8 days similar to their free-floating counterparts. The glucose responsiveness of human donor islets encapsulated inside PVDF/PVP membrane-based devices was significantly improved compared to islets seeded in devices made from PVDF membranes without PVP, with a stimulation index of 3.2 for PVDF/PVP devices and 1.3 for PVDF-alone devices at day 8. Our data show that by addition of PVP as pore forming agent during membrane fabrication at a specific ratio the diffusion characteristics can be tuned such that human islet function in these closed macrodevices, can be kept at the same level as non-encapsulated islets, while the membrane can still serve as a protective barrier preventing the entry of primary human macrophages and damaging beta cells.

**Data availability statement:** All relevant data are within the manuscript and its Supporting Information files.

**Funding:** This work was funded via Regenerative Medicine Crossing Borders (RegMed XB): a public-private partnership that uses regenerative medicine strategies to cure common chronic diseases. This collaboration project is financed by the Dutch Ministry of Economic Affairs by means of the PPP Allowance made available by the Top Sector Life Sciences & Health to stimulate public-private partnerships. The funders of this study had no role in study design, data collection and analysis, decision to publish, or preparation of the manuscript.

**Competing interests:** Aart van Apeldoorn and Denise de Bont are coauthors on a patent currently in review on an "Immunoprotective type implantable cell delivery device" submitted by Maastricht University. This does not alter our adherence to PLOS ONE policies on sharing data and materials.

## Introduction

Type 1 diabetes (T1D) is an autoimmune disease where the insulin producing beta-cells, located in the islets of Langerhans, are specifically destroyed by the immune system [1–4]. It is estimated that 20 to 40 million people worldwide suffer from T1D [5]. Most T1D patients are prescribed with daily exogenous insulin therapy, however, its dose titration needs to be tightly controlled to prevent hypo- or hyperglycaemic events as these can result in life-threatening complications if left untreated [3,6]. An alternative and a rather curative approach to replace insulin therapy is clinical islet transplantation (CIT).

CIT is a minimally invasive therapy where islets are isolated from a donor pancreas and transplanted into the recipient's hepatic portal vein [7,8]. Despite its success, CIT is only available for patients who suffer from glycaemia unawareness and long-term diabetic complications, due to donor organ shortage and the risks associated with long-life immunosuppressive therapy to prevent alloimmune rejection of the donor cells [9–11]. CIT does not consistently lead to long-lasting insulin independence as 50-70% of the transplanted islets are lost shortly after transplantation, mostly due to an instant blood mediated inflammatory reaction (IBMIR) [10,12–14]. IBMIR is triggered by the direct exposure of islets to blood, resulting in rapid activation and binding of platelets to the islets. This leads to a non-specific inflammatory and thrombotic reaction which triggers the innate immune system, finally leading to loss of the transplanted cells [15,16].

Immunoprotective islet encapsulation devices based on thin membranes for extrahepatic transplantation can circumvent rejection of the allogeneic cells by blocking the interaction with the host immune cells, while still allowing the exchange of nutrients and hormones. In a previous study, we successfully demonstrated the use of a polymer-based cell delivery device equipped with microwells to transplant islets in the epididymal fat pad in rodents [52]. The device scaffold was designed to have large pores (up to 40 μm diameter) to allow vascularization post-implantation. This design, however, does not prevent the infiltration of immune cells thus requires the use of immunosuppressive drugs and the many side effects associated with their use remained a challenge [17–19]. We have shown in a previous study that different biocompatible clinically used biomaterials can induce different degrees of oxidative stress in alpha- and beta-cells [20]. Based on these findings, we selected polyvinylidene fluoride (PVDF) for two important reasons it induces relatively low oxidative stress in beta-cells compared to other clinically relevant biomaterials, and it is already used in hernia meshes in the clinic where it has been shown to induce low foreign body reaction preventing fibrous capsule formation around an implant ensuring good biocompatibility [21]. While PTFE has been used in the past for immunoprotective devices like, Theracyte's encaptra or Viacyte's cell pouch, PVDF in comparison to PTFE are more suited for the filtration of non-aggressive aqueous and mild organic while PTFE membranes are commonly used in the filtration of strong acids and aggressive solutions [44,50,51]. Membranes made from both materials have low protein binding ability, and have good chemical compatibility and are both non degradable polymers. It's difficult to make a direct comparison between membranes made from PVDF or PTFE while they're made in a completely different fashion. Studies on hollow fiber membranes and flat membranes related to water purification and food industry indicate that membranes made from PVDF have a higher wettability and higher water flux which is beneficial for diffusion of water-soluble molecules [48,49]. PVDF has a high chemical resistance, mechanical strength, and thermal stability making it a non-degradable, durable, and inert biomaterial [22,23]. In order to increase the porosity of PVDF membranes made by solvent non solvent casting, a water-soluble pore-forming reagent, polyvinylpyrrolidone (PVP) can be used [24–27]. PVP can be mixed into the dissolved PVDF stock solution, which is then used to create thin

membranes by solvent non-solvent casting, after immersion in water, the water soluble PVP leaches out leaving open pores behind. Here we report on the development of an immunoprotective extrahepatic islet delivery device to overcome the use of immunosuppressive drugs based on tailor-made thin PVDF membranes. We assessed the membranes' mechanical properties and ability to allow efficient mass transport of relevant molecules such as oxygen, insulin, and glucose. The diffusion coefficient of glucose and insulin through PVDF/PVP membranes were respectively four and fivefold higher compared to PVDF-alone membranes. Immune cells were not able to migrate through the pores present in the created membranes, both the PVDF-alone and PVDF/PVP membranes. Finally, oval-shaped devices were assembled from the optimized PVDF-alone and PVDF/PVP membranes to evaluate the function and survival of primary human donor islets encapsulated inside these devices. We showed that primary human islets remained viable and functional for 8 days, like free-floating controls.

## Materials and methods

### Porous PVDF membrane production

Casting solutions were made by dissolving 15% (w/w) polyvinylidene fluoride (PVDF) (Kynar 720, Foster, Germany) in N, N-dimethylacetamine (DMA) (Sigma-Aldrich, the Netherlands). Solutions were mixed overnight at room temperature (RT) on a roller bank until a homogeneous solution was obtained without air bubbles. For polyvinylpyrrolidone (PVP) containing membranes, 5% (w/w) PVP (40 kDa) (Kollidon® 30, BTC, the Netherlands) was added to the PVDF/DMA solution after overnight mixing, and left on the roller bank for another 24 h to obtain a homogeneous solution. Solutions were then casted on glass plates using an automatic film applicator (Elcometer, the Netherlands) equipped with a 200 μm high casting knife (at RT and relative humidity < 30%), immediately followed by direct immersion of the glass plates into a coagulation bath (distilled water at RT). Membranes were collected from the coagulation bath and dried between paper tissues for two days before further use.

### Scanning electron microscopy (SEM)

Membrane samples were fixed on metal SEM stubs using carbon tape and gold coated using a sputter coater (Cressington, United Kingdom). Cross-sections of membranes were made by immersing membranes in liquid nitrogen prior to breaking each sample in half prior to imaging. Electron micrographs were taken using a scanning electron microscopy (SEM) (Teneo, FEI, the Netherlands) under high vacuum using secondary electron mode with ETD detection, 10 mm working distance and 5.00 kV acceleration voltage. Average pore sizes in membranes were determined by using the SEM's measurement software tool, to measure the dimensions of at least 10 randomly selected pores on each membrane.

### Support ring production

PVDF pellets were preheated between a metal mold (200 μm thick, made in house) inside a hot press (Specac, United Kingdom) at 180 °C for 1 minute. Next, 20 kN was applied to soften pellets for 1 minute to create a solid film. Finally, the metal mold was removed from the hot press and cooled down for 5 minutes at RT before detaching the solid PVDF film from the mold.

### Membrane wettability

Membranes were mounted horizontally on a glass slide. Droplets of 4 μL of distilled water (at RT) were dispensed on the surface of the membrane at a speed of 10 μL/min and high-resolution images were taken. Contact angles were measured by using Drop Shape Analysis

4 software equipped with a drop shape analyser (Krüss, Germany). Measurements of nine droplets were taken for each membrane.

## Mechanical tensile testing

Mechanical tensile properties of produced films and sealing methods were determined by mechanical tensile testing instrument (45 kN load cell, Electroforce 3230-ES Series III, United Kingdom). Dimensions of test samples were 35 x 10 mm, with an effective area of 15 x 10 mm between the clamps. The ramp speed was 0.05 mm/s. Each condition was tested five times. Obtained stress-strain curves were used to calculate peak strain, failure stress, failure strain and Young's modulus. The different device components were annealed to each other using two different methods for comparison: heat sealing by a flat wire impulse sealer (Durapak, United States) and ultrasonic welding (LPX, Branson Ultrasonics, the Netherlands). Subsequently, the strength of the seals was determined with mechanical tensile.

## Cell culture

INS-1E rat insulinoma cells (Addexbio Technology, United States) were cultured in Roswell Park Memorial Institute (RPMI) medium with L-glutamine (Sigma-Aldrich, the Netherlands) supplemented with 10% (v/v) fetal bovine serum (FBS) (Sigma-Aldrich, the Netherlands), 10 mM HEPES (Thermo Fisher Scientific, the Netherlands), 1 mM sodium pyruvate (Thermo Fisher Scientific, the Netherlands), 5 mM glucose (Thermo Fisher Scientific, the Netherlands), 23.8 mM sodium bicarbonate (Thermo Fisher Scientific, the Netherlands) and 50 mM beta-mercaptoethanol (Thermo Fisher Scientific, the Netherlands). INS-1E P38 till P43 as a model for beta cells were used. Human primary macrophages (Celprogen, United States) were cultured in Dulbecco's Modified Eagle Medium (DMEM) high glucose (Thermo Fisher Scientific, the Netherlands) supplemented with 10% (v/v) FBS (Sigma-Aldrich, the Netherlands). Macrophages P6 till P10 were used. All cells were cultured at 37°C and 5% $CO_2$.

## Human islet culture

Human donor islets of Langerhans were kindly provided by the islet isolation and transplantation center at the Leiden university medical center (LUMC, Leiden, the Netherlands). Human islets were obtained from pancreata retrieved from braindead donors who previously registered as organ donors, and when not deemed suitable for clinical transplantation were used in these experiments in accordance with Dutch law. None of the transplant donors was from a vulnerable population and informed consent was obtained from all subjects and/or their legal guardian(s) prior to organ retrieval, while no financial transaction was performed for the obtained tissue in accordance with Dutch law. Donor tissue was provided in an anonymized way. Experiments performed with human islets in this paper were done in accordance with the relevant guidelines and regulations for research purposes. One islet equivalent (IEQ) is regarded as an islet of having a diameter of 150 μm, typically containing 1,500-2,000 cells, including 40-60% beta cells. Islets of different sizes are volumetrically adjusted to IEQs during seeding in free floating and device conditions. Organ donors (2 males and 1 donor gender unknown) had an average age of 43 ± 14 years. The average islet purity was ~75 ± 5%. Islet purity is determined in the by a standard Dithiazone staining and microscopic evaluation. Dithiazone stains zinc associated to insulin red, which can be used to distinguish insulin producing islets from non-endocrine tissue in isolation preps. Islets were cultured in CMRL-1066 medium (Pan Biotech, Germany) supplemented with 10% (v/v) FBS (Sigma-Aldrich, the Netherlands), 10 mM HEPES (Thermo Fisher Scientific, the Netherlands), 1% penicillin-streptomycin (Thermo Fisher Scientific,

the Netherlands) and 10 μg/mL ciprofloxacin (Sigma-Aldrich, the Netherlands). Islets were cultured at 37°C and 5% $CO_2$.

## MTT assay

We performed a cytotoxicity study similar to what is usually done with fibroblasts for biomaterials testing described in the ISO 10993-5: MTT & MEM Elution Test. The cellular metabolic activity was assessed by measuring the conversion of 3-(4,5-dimethylthiazol-2-yl)-2,5-diphenyltetrazolium bromide (MTT) into formazan. This process only takes place in metabolic active cells. Medium extraction samples were generated by 24h incubation of test materials in RPMI cell culture medium (see section cell culture) to collect any potential cytotoxic leachable eluting from devices. 1% Triton X-100 (VWR, the Netherlands) in RPMI was used as negative control inducing cytotoxicity, while untreated medium is used as positive control. INS-1E cells were seeded with $5 \times 10^5$ cells/mL in 96-well plate (100 μL) and incubated overnight to form a 50% confluent monolayer in RPMI medium. After 24h medium was removed and 100 μL extract sample was added and incubated overnight. After 24h the medium was replaced by 100 μL of fresh cell culture medium supplemented with 10 μL of 12 mM MTT (Thermo Fisher Scientific, the Netherlands) solution in phosphate buffered saline (PBS) (Sigma-Aldrich, the Netherlands) and incubated for 4h. All medium was removed except 25 μL mixed with 100 μL of DMSO (VWR, the Netherlands) and incubated for 10 minutes at 37°C. Absorbance was measured with CLARIOstar microplate reader (BMG Labtech, Germany) at 540 nm. Cell viability was calculated with equation (1) ($OD_{540e}$ = measured optical density of extracts of test sample, $OD_{540b}$ = measured optical density of blank).

$$Viability(\%) = \frac{100 * OD_{540e}}{OD_{540b}} \tag{1}$$

## Lactate dehydrogenase (LDH) cytotoxicity assay

Cell cytotoxicity was assessed by measuring by the LDH levels in cell culture medium. INS-1E cells were seeded with $5 \times 10^5$ cells/mL in a 96-well (100 μL) and incubated overnight. Next, 10 μL extract sample was added (see MTT assay) and incubated for 45 minutes. 50 μL of each sample was transferred to a 96-well plate and 50 μL reaction mixture (Pierce LDH, Thermo Fisher Scientific, the Netherlands) was added and incubated for 30 minutes at RT while protected against light. 50 μL of stop solution was then added and mixed, followed by measuring absorbance at 490 nm and 680 nm with CLARIOstar microplate reader. Cytotoxicity was calculated with equation (2).

$$Cytotoxicty(\%) = \frac{Compound\ treated\ LDH\ activity - Spontaneous\ LDH\ activity}{Maximum\ LDH\ activity - Spontaneous\ LDH\ activity} * 100 \tag{2}$$

## Fourier-transform infrared spectroscopy (FTIR)

Fourier transform infrared spectroscopy (FTIR) (Nicolet iS50-FT-IR, Thermo Scientific, the Netherlands) was used to analyse the chemical composition of the produced membranes. Infrared spectra were recorded at a wavelength range of 400–4000 cm$^{-1}$. The results were analysed using SpectraGryph (version 1.2.13, Germany) to identify peaks related to chemical bonds.

## Assembly of encapsulation devices

Membranes and support rings were cut using a cutting machine (Curio, Silhouette, the Netherlands). The membranes are shaped into an oval (44 x 28 mm) and the oval support ring

(44 x 28 mm) has a width of 4 mm and contains a 6 mm wide opening. Devices were produced by placing a support ring between two membranes sheets and annealing them by ultrasonic welding (LPX, Branson Ultrasonics, the Netherlands) inside a custom-made mold (IDEE, the Netherlands). The assembled devices were tested for leakage by an air bubble test. Briefly, assembled devices were submerged in water while air (~4 L/min) was blown through the inlet into the device. Leakages were then detected when escaping air bubbles from the device were observed

## Membrane diffusion properties

Membranes were pre-wetted for 2 min in 70% ethanol followed by washing for 2 min with distilled water to remove ethanol. Wetted membranes were placed in In-Line Equilibrium diffusion cells (Sigma-Aldrich, the Netherlands) containing a donor and a receiver chamber. The donor chamber was filled with a PBS solution containing either 2 mg/mL 3-5 kDa FITC-dextran (Sigma-Aldrich, the Netherlands), a 20 mM D-glucose solution (Sigma-Aldrich, the Netherlands), a 0.5 mg/mL FITC-labelled human insulin (Sigma-Aldrich, the Netherlands), or a NaCl solution (150 mM) containing 1 mg/mL IgG (Sigma-Aldrich, the Netherlands). The receiver chamber was filled with PBS for dextran, glucose, or insulin diffusion; or 150 mM NaCl for IgG diffusion. Ten µl fluid was sampled from both chambers at 0, 5, 10, 20, 30, 60, 120, 180 minutes, in the case of insulin diffusion an additional sample at 1440 minutes was taken. Quantitative analysis of diffused FITC-dextran and FITC-insulin was determined based on fluorescent intensity using a CLARIOstar microplate reader (excitation and emission wavelength are 480nm and 530 nm, respectively). The amount of glucose was measured using a colorimetric glucose assay (Bio-connect Diagnostics, the Netherlands) according to the manufacturer's protocol. Briefly, samples were collected and diluted in PBS. 50 µL of glucose solution was incubated with 50 µL of reaction mix for 20 minutes at 37°C. Absorbance was measured using the CLARIOstar microplate reader (excitation wavelength = 555 nm). The amount of IgG was measured with an IgG ELISA (Thermo Fisher Scientific, the Netherlands) according to the manufacturer's protocol. The flux values were calculated using equation (3) to gain insight into how much of a molecule of interest was diffused over the membrane at a specific time ($C_{Receiver}$ = concentration at receiver chamber, $V_{Receiver}$ = volume of receiver chamber, A = area of membrane, $t$ = time). The diffusion coefficient was calculated using equation (4) ($\Delta C$ = concentration difference between donor and receiver, $l$ = thickness of membrane). Each condition was tested at least four times.

$$flux\left(gm^{-2}s^{-1}\right) = \frac{C_{Receiver}\left(gm^{-3}\right) * \dfrac{V_{Receiver}\left(m^{3}\right)}{A\left(m^{2}\right)}}{t\left(s\right)} \tag{3}$$

$$diffusion\ coeffient = \frac{flux\left(gm^{-2}s^{-1}\right)}{\Delta C\left(gm^{-2}\right)} * l\left(m\right) \tag{4}$$

## Oxygen diffusion properties inside the device

Before the devices were loaded with medium, the assembled devices were pre-wetted for 2 min in 70% ethanol, followed by 2 min washing with distilled water to remove ethanol. Oxygen diffusion was measured by loading the device with 500 µL of RPMI cell culture medium. Medium loaded devices were heat-sealed and placed inside a 100 mL beaker containing 100 mL of fresh cell culture medium. The beaker containing the device was then placed at

37°C in a humidified incubator with 5% $O_2$. The oxygen concentration within the devices and the outside were recorded for 24 h, using a previously calibrated needle-type oxygen micro-sensor (NTH-PSt7, Presens, Germany) connected to an oxygen meter (OXY-10, Presens, Germany). Devices were kept submerged in media during the entire oxygen measurement duration. Each condition, free-floating controls (islets cultured on a non-adherend tissue culture plate), PVDF-alone and PVDF/PVP devices, were tested three times independently.

## Immunoprotective capacity of membranes

Membranes were mounted onto 0.33 cm$^2$ empty transwell inserts (Corning, the Netherlands) with glue (Bison, the Netherlands) and O-rings (Eriks, the Netherlands). After allowing the glue to dry overnight, inserts were sterilized in 70% ethanol. Human primary macrophages were seeded (1.0 x 10$^5$ cells in 100 μL) onto the top compartment of each insert in a 24-well plate, 600 μL medium was added to each well. Cells were cultured for 24 h (37 °C, 5% $CO_2$). Cells which migrated to the bottom compartment were counted manually using Neubauer haemocytometer. To observe cell adherence and infiltration through the membranes, membranes were fixed with 3.6% formaldehyde (VWR, the Netherlands) in PBS for 30 minutes at RT, followed by multiple rinses in fresh PBS, membranes were then cut from the inserts. Samples were dehydrated with increasing ethanol concentrations (30, 50, 70, 90, 96, 100 and 100% ethanol) (VWR, the Netherlands) for 10 minutes each. Dehydrated samples were processed using an automated critical point dryer (Leica, the Netherlands). Subsequently, the dried membrane samples were prepared and imaged using scanning electron microscopy as described previously.

## Islet seeding into the encapsulation devices

Devices were disinfected by overnight incubation in 70% ethanol, followed by washing with sterile PBS. Before seeding islets into the devices, devices were incubated in medium overnight (37°C, 5% $CO_2$). 3,000 human islet equivalents (IEQ) in 500 μL medium was seeded inside each device. For the control samples, 30 IEQ were seeded in 12 μm cell culture inserts (Merck, the Netherlands). The seeding inlet was closed with heat sealing (Durapak, United States). Devices were cultured for 8 days (37°C, 5% $CO_2$) and medium was refreshed every other day.

## Human islet viability

The viability of human islets was assessed using live-dead fluorescent dyes, calcein AM and ethidium homodimer-1 (Thermo Fisher Scientific, the Netherlands) for live and dead cells, respectively. After 8 days of culturing, devices were opened, and the islets were flushed out of the device using PBS. The islets were then incubated with 2.5 μM calcein AM and 4 μM ethidium homodimer-1 in PBS for 30 minutes (while protected against light) and visualized under fluorescent microscopy (Eclipse Ti-S inverted microscope, Nikon, the Netherlands). The percentage of viability was quantified by dividing the area of live cells by the total area of cells (live and dead).

## Human islet function during glucose stimulation

To assess the insulin release of human islets inside the devices a glucose stimulated insulin secretion (GSIS) assay was performed. Krebs buffer was prepared by dissolving 25 mM HEPES (Sigma-Aldrich, the Netherlands), 115 mM sodium chloride (Sigma-Aldrich, the Netherlands), 26 mM sodium bicarbonate (Sigma-Aldrich, the Netherlands), 5 mM potassium chloride (Sigma-Aldrich, the Netherlands), 1 mM magnesium chloride dihydrate (Sigma-Aldrich, the Netherlands), 25 mM calcium chloride dihydrate (Sigma-Aldrich, the Netherlands) with

2% bovine serum albumin (BSA) (VWR, the Netherlands) in MilliQ at pH 7.3-7.5 and filtered sterile. Low and high glucose solutions contained 1.67 and 16.7 mM D-glucose in Krebs buffer respectively. After 8 days of culturing, devices and islets were washed three times in low glucose solution, followed by 1 h pre-incubation in low glucose (37°C, 5% $CO_2$). Next, devices and islets were exposed to low – high – low glucose solutions for 1 h each (37°C, 5% $CO_2$) after which supernatant was collected and stored at -30°C until analysis was performed. Devices and islets were washed three times with low glucose solution after high glucose exposure. After the final incubation step, devices were opened and islets were exposed to acid ethanol to release all insulin inside the cells. Insulin secretion was determined with human insulin ELISA kit (Mercodia, Sweden) according to manufacturer's protocol. In order to compensate for any differences used in IEQs used per experimental condition Free floating vs device insulin secretion is depicted as insulin secreted in the medium relative to total insulin present in all beta cells. To determine total insulin cell were lysed after each experiment after which ELISA was used to determine the total amount of insulin. The stimulation index was calculated by dividing the insulin secretion in high glucose solution by the basal insulin secretion in low glucose solution.

## Data analysis

Results are presented as mean ± standard deviation (SD), unless otherwise specified. Experiments were performed in triplicate unless specified. For each experiment three samples were taken per experimental condition. Statistical analysis was performed by using GraphPad Prism 9 (GraphPad Software, United States). Statistical difference between two groups was assessed using an unpaired student T-test. Statistical differences between multiple groups were assessed using a Kruskal-Wallis when the data were not normally distributed, or a one-way ANOVA test with multiple comparison when the data were normally distributed. Statistical significance was considered when $p < 0.05$.

## Results

### Fabrication and physicochemical characterization of membranes

Thin and porous membranes were made using a solvent non-solvent casting method, by casting 15% (w/w) PVDF + 5% (w/w) PVP (PVDF/PVP) and 15% (w/w) PVDF (PVDF-alone) solutions onto glass plates. The surfaces of both PVDF membranes, with and without PVP, have a smooth and uniform appearance. Larger pores were observed on the glass-side (Fig 1A and 1D) compared to the air-side (Fig 1B and 1E) of the membranes. The glass-side pores in PVDF/PVP membranes had an average diameter of 0.51 ± 0.15 μm, while the glass-side pores in PVDF-alone membranes had an average diameter of 0.25 ± 0.12 μm (Fig 1G). The presence of PVP increased the average pore size on the air-exposed side (Fig 1B and 1E), with a mean diameter of 113 ± 4 nm compared to PVDF-alone membranes (51 ± 2 nm) (Fig 1H). The addition of PVP resulted in a two-fold increase in the pore size diameter both on the glass- and air-sides. Observations of cross-sections reveal that the appearance of PVDF-alone membranes was denser and thinner (45 ± 4 μm), compared to PVDF/PVP membranes (137 ± 19 μm) (Fig 1C, 1F and 1I).

Contact angle measurements were performed by static sessile drop method to determine the hydrophilic/hydrophobic properties of produced membranes. Unlike that of the glass-exposed side (both approximately 73°), the surface wettability of the air-exposed side of the PVDF/PVP membranes was significantly lower compared to PVDF-alone membranes (80 ± 3° vs 99 ± 4°; Fig 2A and 2B). The mechanical properties of the produced membranes were determined with mechanical tensile testing. PVDF/PVP membranes showed significantly

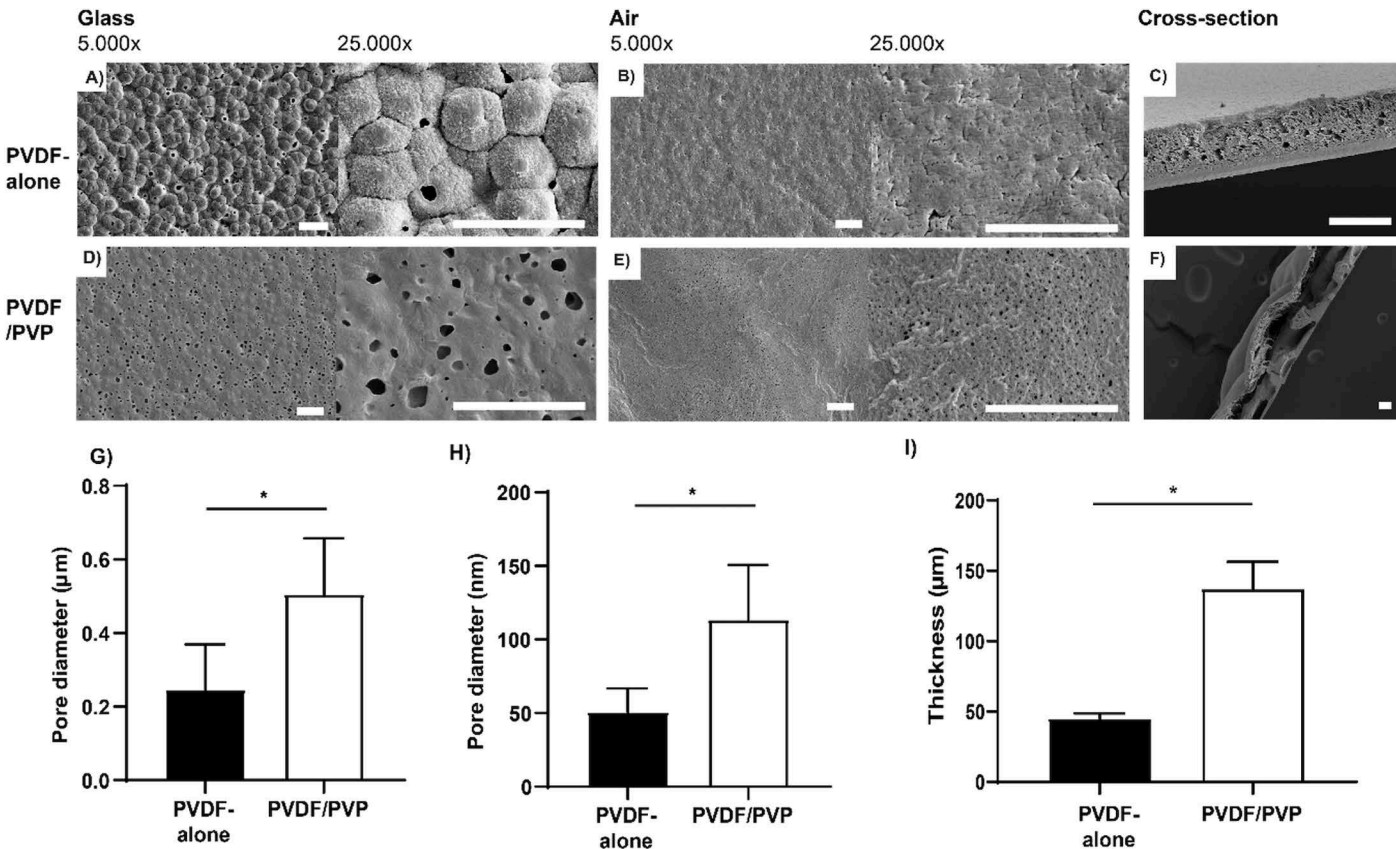

**Fig 1. Representative SEM images of surface morphology and cross-section of produced PVDF-alone and PVDF/PVP membranes.** 15% (w/w) PVDF membrane casted at 200 μm (A–C) and 15% (w/w) PVDF + 5% (w/w) PVP membranes casted at 200 μm (D–F). Views of the glass-side at 5.000× and 25.000× magnification (A, D); air-side at 5.000× and 25.000× magnification (B, E); and cross-section (C, F). Scale bars in A, B, D and E represents 4 μm. Scale bar in C and F represents 50 μm. (G, H) Pore diameter of produced membranes measured at the glass- (G) and air-exposed side (H). Data presented as mean ± SD, n = 3. (I) The thickness of produced membranes measured from SEM cross-section images. *P < 0.05, with *t*-test.

less mechanical strength compared to PVDF-alone membranes as determined by the Young's modulus (0.32 ± 0.04 MPa for PVDF/PVP and 0.87 ± 0.11 for PVDF-alone; Fig 2C) and peak stress measurements (0.97 ± 0.03 MPa for PVDF/PVP and 4.95 ± 0.16 for PVDF-alone; Fig 2D). Moreover, both failure strain (Fig 2E) and the failure stress (Fig 2F) were significantly lower in PVDF/PVP membranes compared to PVDF-alone membranes: 0.92 ± 0.06 MPa vs 4.78 ± 0.07 MPa and 11 ± 1% vs 84 ± 8%, respectively.

## Cytotoxicity of membranes

To determine the cytotoxicity of produced PVDF membranes and support rings, LDH and MTT assays were performed similar to the method described in ISO10993-5: MTT & MEM elution test using ISN1E cells. Cell viability was significantly lower in PVDF/PVP membranes compared to PVDF-alone membranes (92 ± 8 vs 106 ± 9%; Fig 3A). Neither membrane showed significant differences in cytotoxicity compared to the untreated cells as positive control (-7 ± 15 vs 10 ± 14% for PVDF-alone and PVDF/PVP membranes respectively; Fig 3B). Although PVP is water-soluble, residual PVP particles may have remained after membrane fabrication that influenced our cytotoxicity measurements. Therefore, FTIR analysis was performed to determine if there were still PVP residue present (Fig 3C). In PVDF-alone

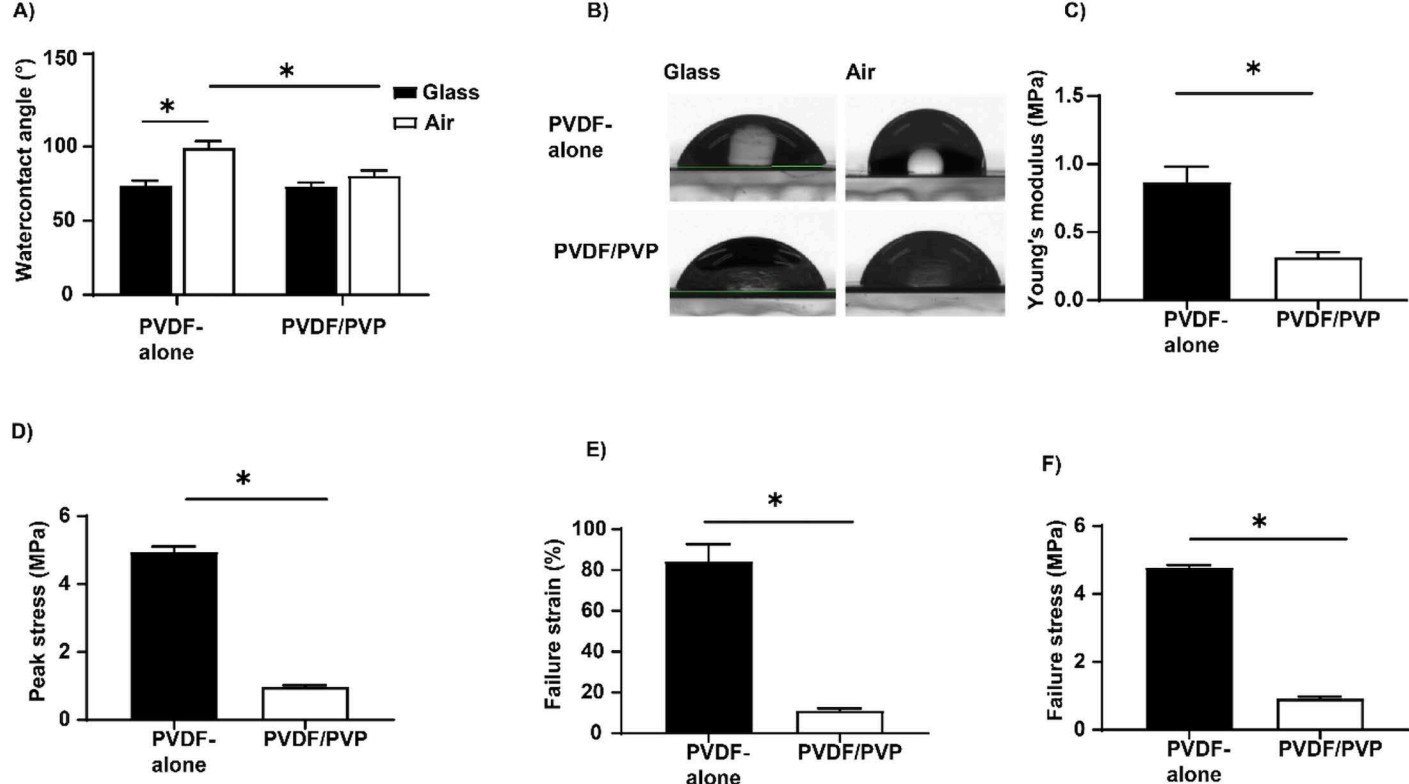

**Fig 2. Physical properties of the produced PVDF-alone and PVDF/PVP membranes.** **(A)** Water contact angle from both sides of the produced membranes. Data presented as mean ± SD, n = 9. * P < 0.05, with student t-test. **(B)** Representative images of water contact angle measurements of the produced membranes. **(C–F)** Mechanical properties of produced membranes, characterized by the Young's modulus **(C)**, peak stress **(D)**, failure strain **(E)**, failure stress **(F)**. Data presented as mean ± SD, n = 5. * P < 0.05, with student t-test.

membranes, peaks were present at ~ 1400 cm$^{-1}$ and ~ 1180 cm$^{-1}$, corresponding to $CH_2$ and $CF_2$ stretching groups present in the base structure of PVDF. In PVDF/PVP, an additional peak at ~ 1660 cm$^{-1}$ was observed, corresponding to the C = O group present in PVP (green dotted line in Fig 3C). PVDF/PVP membranes are slightly more cytotoxic to INS-1E cells, which can be attributed to residual PVP particles.

## Diffusion of small molecules through membranes

To gain insight into the transport of different molecules across membranes, we tested four molecules; dextran, glucose, insulin, and IgG. The membranes were placed in-between two compartments, a donor and a receiver chamber. The concentration measurements of the different molecules were taken from each chamber at predefined time points to calculate the diffusion kinetics. For all four molecules, diffusion was greater and faster in PVDF/PVP membranes compared to PVDF-alone membranes. The diffusion of dextran (3–5 kDa) through PVDF/PVP was nearly three-fold higher than in PVDF-alone membranes (Fig 4A) and the calculated flux and diffusion coefficient were significantly higher (Fig 4B and 4C). Similarly, glucose molecules diffused faster through PVDF/PVP membranes, reaching equilibrium (equal in the two chambers) after 3 h, in contrast to only 30% of the glucose diffusing through PVDF-alone membranes (Fig 4D) at the same time point. This observation was confirmed by the significantly higher glucose flux of 10 x 10$^{-4}$ g m$^{-2}$ sec$^{-1}$ and diffusion coefficient of 4.0 x 10$^{-7}$ cm$^2$ s$^{-1}$ for the PVDF/PVP membrane compared to PVDF-alone (4 x 10$^{-4}$ g m$^{-2}$ sec$^{-1}$ and

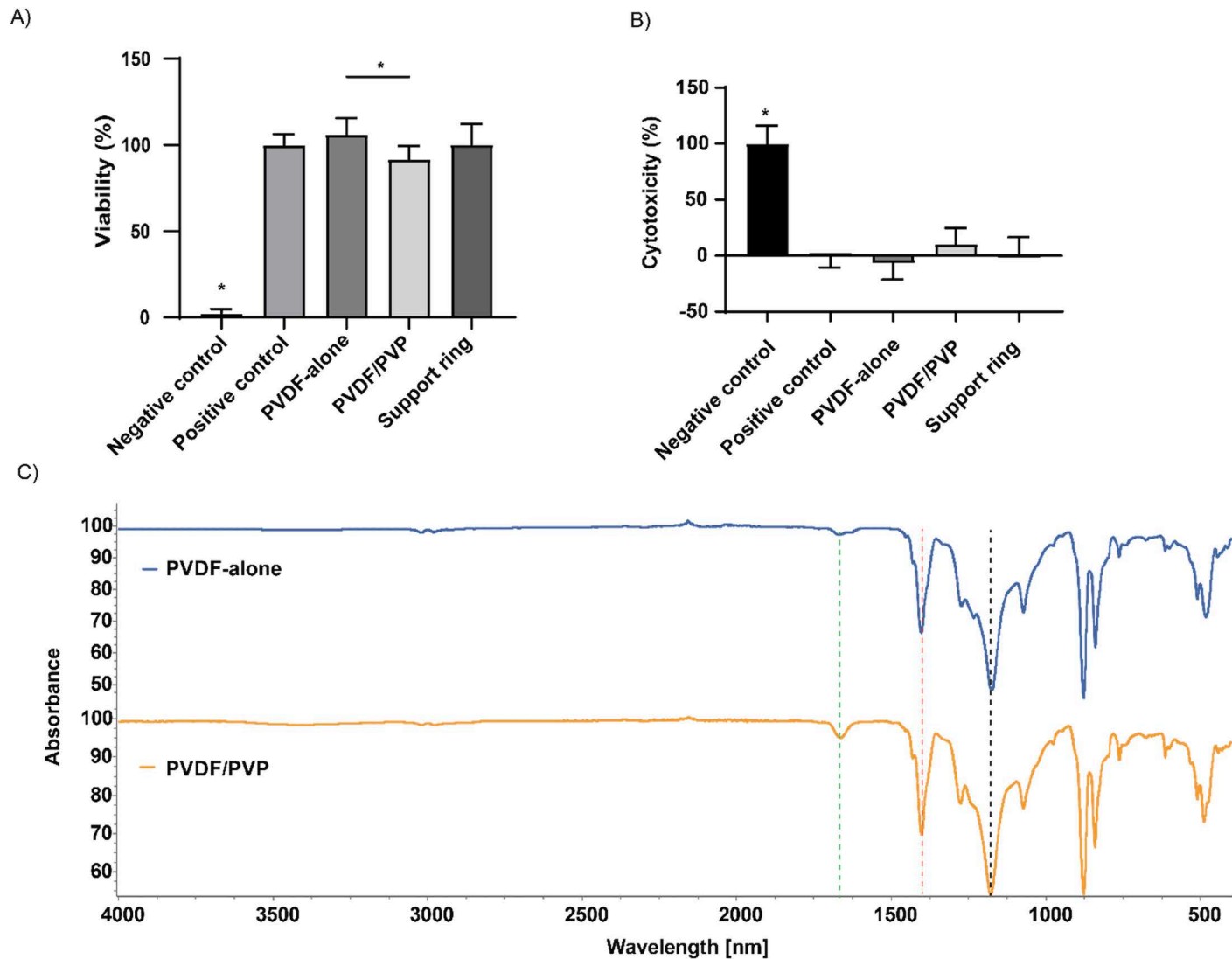

**Fig 3. Indirect cytotoxicity tests and chemical composition of the produced PVDF-alone and PVDF/PVP membranes. (A)** Levels of metabolic activity of INS-1E cells in contact with different exposure media during a MTT assay. Data presented as mean ± SD, n = 3, *P < 0.05, one-way ANOVA. **(B)** Measurement of LDH release of INS-1E cells in contact with different exposure media. Data presented as mean ± SD, n = 3, *P < 0.05, one-way ANOVA. **(C)** FTIR spectra of produced membranes. The green line at ~ 1660 cm$^{-1}$ indicate a C=O peak present in PVDF/PVP membranes. The red and black lines at ~ 1400 and ~ 1180 cm$^{-1}$ respectively correspond to CH$_2$ and CF$_2$ stretching groups present in the PVDF base structure.

1.1 x 10$^{-7}$ cm$^2$ s$^{-1}$, respectively; Fig 4E and 4F). Glucose was not absorbed by the membrane, as the total concentration of glucose remained the same over time. FITC-labelled insulin reached equilibrium after 24 h in PVDF/PVP membranes as 47 ± 2% of insulin was diffused, compared to whereas 28 ± 5% was diffused in PVDF-alone membranes at the same time point (Fig 4G). The insulin diffusion capacity was confirmed by significantly higher of 1.91 x 10$^{-5}$ g m$^{-2}$ sec$^{-1}$ and diffusion coefficient of 1.31 x 10$^{-8}$ cm$^2$ s$^{-1}$ in PVDF/PVP membranes compared to PVDF-alone membranes (0.35 x 10$^{-5}$ g m$^{-2}$ sec$^{-1}$ and 0.08 x 10$^{-8}$ cm$^2$ s$^{-1}$, respectively; Fig 4H and 4I). For IgG only 1.6 ± 0.7% of the total amount was able to diffuse through PVDF/PVP membranes, while no IgG could be detected for PVDF-alone membranes, indicating that the increased porosity of PVDF/PVP membranes does allow for some IgG to pass these membranes (Fig 4J). Because of the low levels of diffusion, the flux and diffusion coefficient for IgG

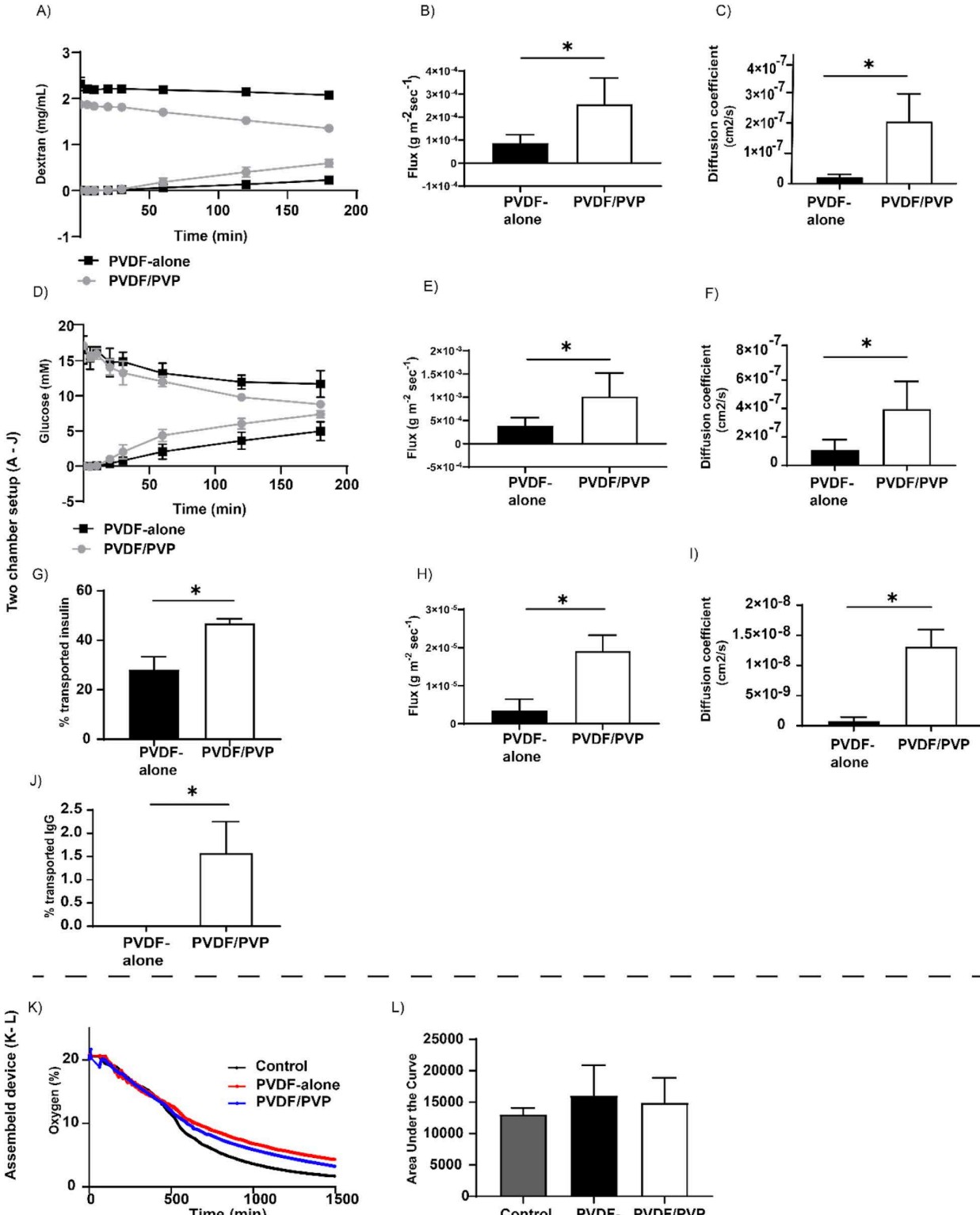

**Fig 4. Diffusion properties of different molecules over the produced PVDF-alone and PVDF/PVP membranes, either in 2-chamber diffusion setup (A–J) or in assembled devices (K–L).** (A–C) Diffusion of 2 mg/mL of 3-5 kDa fluorescently labelled dextran bead over 3 h (**A**), its calculated flux (**B**) and diffusion coefficient (**C**). Data presented as mean ± SD, $n = 5$. $*P < 0.05$, with student $t$-test. (**D–F**) Diffusion of 20 mM D-glucose over 3h (**D**), its calculated flux (**E**) and diffusion coefficient (**F**). Data presented as mean ± SD, $n = 5$. $*P < 0.05$, with student $t$-test. (G–I) Diffusion of FITC-labelled insulin, as a percentage of total after 24h (**G**), its calculated flux (**H**), and diffusion coefficient (**I**). Data

presented as mean ± SD, $n = 5$. *$P < 0.05$, with student t-test. (**J**) Diffusion of IgG, as a percentage of total after 3 h. Data presented as mean ± SD, $n = 4$. (**K**–**L**) Diffusion of oxygen from 21% oxygen towards 5% oxygen (**K**) and *area under the curve* (*AUC*) of oxygen diffusion (**L**). Data presented as mean ± SD, $n = 3$.

could not be calculated. The transport of oxygen was measured by filling the devices with oxygenated medium (21% oxygen) and the filled devices were placed in medium inside a hypoxia incubator (5% oxygen). The oxygen concentration inside the devices was measured over time by placing an oxygen probe inside the device. The transport of oxygen was not significantly different in free solution (cell culture medium) compared to both PVDF-alone and PVDF/PVP based membrane devices, as areas under the oxygen curves were similar (Fig 4K–4L).

### Cell barrier protective function of PVDF membranes

To evaluate the protective properties of the tailor-made membranes (PVDF-alone and PVDF/PVP), we assessed the migration of macrophages through the different membranes and compared these to commercially available membranes with specific pore sizes. As shown by scanning electron microscopy (Fig 5A), no macrophages were detected on the bottom side of PVDF-alone and PVDF/PVP membranes, as well as the commercial membrane with 0.4 μm pores serving as a negative control. Macrophages were detected on both sides of commercially available inserts with 3 and 8 μm pores (Fig 5A). Macrophages were detected on the well plate's surface underneath the 8 μm inserts (Fig 5B and 5C). The nuclei of the human macrophages were not able to migrate through 1 μm pores; however, these pores were large enough to enable the migration of filopodia (Fig 5A).

### Assembly of the macroencapsulation devices

To assemble the encapsulation devices, two membrane sheets were welded together with high-frequency ultrasonic welding or heat sealing using a support ring in-between as illustrated in Fig 6A. A small opening was not welded to serve as an inlet for loading cells into the device (Fig 6D). Each device was tested for leakages, using a tailored pressurized air bubble test. Any defects were detected by visually inspecting the escape of air bubbles during the pressure test (S4 Fig D). Devices with welding defects were not used for further experiments.

### Viability and function of human islets inside the encapsulation devices

Human islets were cultured for 8 days inside closed devices to assess viability and function using live-dead staining and a GSIS test. No significant difference in viability was observed between islets cultured inside the PVDF-alone and PVDF/PVP devices, compared to free floating controls (Fig 7A and 7B). The viability of human islets was 92 ± 4% after 8 days of culturing inside PVDF/PVP encapsulation devices. In the control condition, islets responded to changing glucose levels after 8 days as we observed a stimulation index of 2.3 ± 0.4 (Fig 7C and 7D). Islets encapsulated in PVDF/PVP devices responded well to changing glucose levels (Fig 7C). The stimulation index of islets in PVDF/PVP devices was significantly higher than in PVDF-alone devices (3.2 ± 0.7 vs 1.3 ± 0.2) (Fig 7D) indicating PVDF/PVP membranes are more permissive for glucose and insulin exchange.

### Discussion

Here, we report on the development of a closed islet encapsulation device made of microporous PVDF membranes. Immunoprotective beta cell delivery devices can help overcome the challenges faced in clinical islet transplantation therapy by omitting the use of

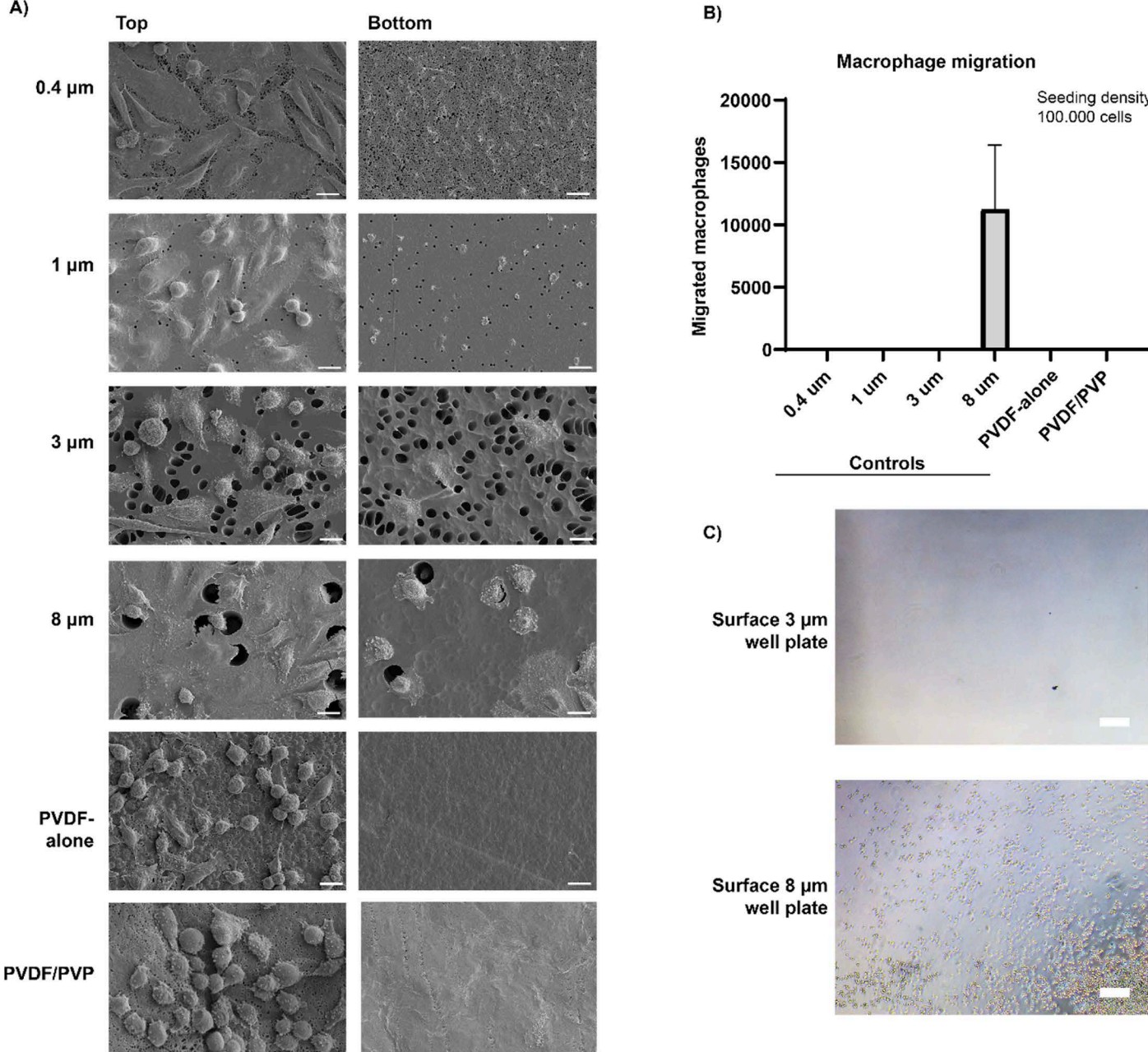

**Fig 5. Immunoprotective properties of produced membranes, PVDF-alone and PVDF/PVP. (A)** Representative SEM images of fixed human primary macrophages seeded on top of different porous transwell inserts (0.4, 1, 3 and 8 μm pores) or produced membranes. Cells that were able to migrate through the pores can be seen in the representative bottom image. Scale bar represents 10 μm in all images **(B)** Quantification of migrated human primary macrophages on the bottom of 24-well plate. Data presented as mean ± SD, n = 3. **(C)** Representative light microscopy images of well plate bottoms of 3 and 8 μm transwell after 24 h. Scale bar represents 200 μm.

immunosuppressive drugs to avoid rejection in patients subject to an islet alone intervention. Those drugs are used to avoid rejection of the allogeneic cells, and to some extent inhibit the autoimmune reaction to beta cells. However, in the same time they can have a detrimental effect on beta cell survival at sufficiently high concentrations, and also lead to unwanted side effects in patients due to dampening of their immune system. Here we report on the design,

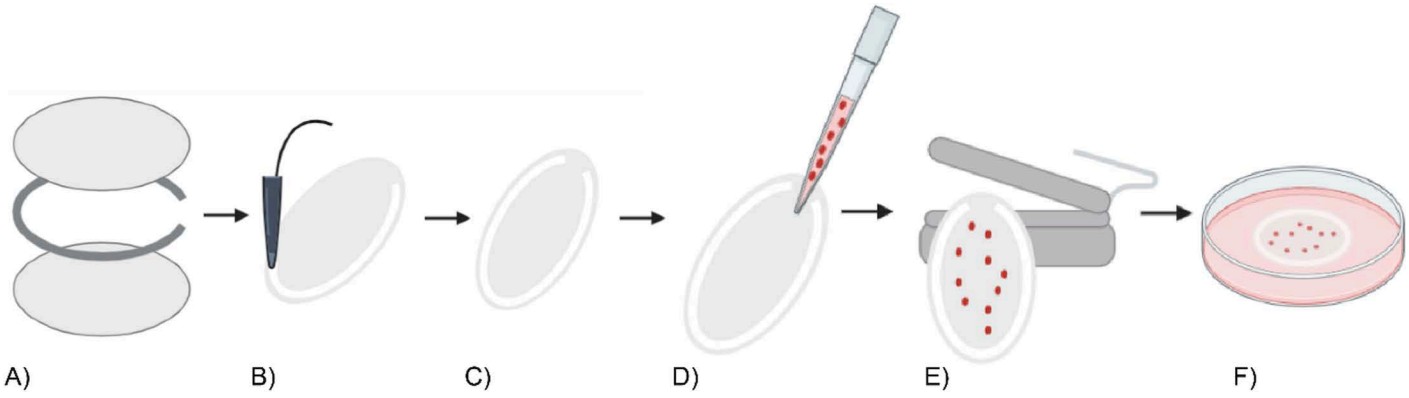

**Fig 6. Schematic overview of device assembly, cell seeding and sealing.** (A) Three layers of the device, two membrane sheets (light gray) and the support ring (dark gray) in between. (B) Sealing the three layers of the device with ultrasonic welding. (C) Assembled device with a seeding inlet. (D) Cell seeding through the inlet. (E) Closing the seeding inlet with heat sealing. (F) Culture of the cell loaded device.

manufacturing and characterization of a thin closed macroencapsulation device based on tailor made PVDF membranes. We show that the pore size in solvent-non solvent casted PVDF membranes is small enough to block immune cell infiltration, and simultaneously allows for sufficient nutrient and insulin diffusion to, and from encapsulated human donor islets when used in a closed cell delivery device, assembled from these membranes.

There are three main challenges in the current development of macroencapsulation devices for islet and beta cells. The first being the biocompatibility of the biomaterial used for cell encapsulation, even though biomaterials might be considered to be biocompatible for a certain specific application, they still can induce unwanted cell stress, and trigger an undesired foreign body response after implantation, when used in another application. We previously published that films made from PVDF are not only strong and flexible, and thus can be used for the creation of thin film cell delivery devices, but importantly PVDF also induces relatively low amounts of oxidative stress in beta-cells, which have an inherently low antioxidant capacity [20]. In addition, PVDF has a long-standing positive track record as a biomaterial for internal clinical use in hernia meshes and suture threads, due to its strength and low foreign body reaction inducing properties making it potentially a suitable material for islets and beta cells delivery devices [21]. The other two challenges are immune evasion, and sufficient mass transport capabilities of the device to support the function and survival of beta cells. Several studies have reported that membranes used in macroencapsulation devices for islets should have a pore size below 0.45 μm to be classified as being immunoprotective [28–30]. However, a reduction in pore size can also significantly slow down the diffusion rate of nutrients, and hormones such as insulin essential for proper islet function.

In this study two formulations of PVDF membranes were compared, PVDF-alone and PVDF/PVP, hypothesizing that the addition of a water soluble porogen such as PVP during membrane manufacturing can increasing pore size and hydrophilicity thereby enhancing the diffusion properties, while still maintaining their cell protective barrier function. PVP is a water-soluble polymer, which can increase membrane porosity by increasing the phase separation in a dissolved PVDF solution when applied in a solvent non-solvent casting technique to create thin films [24–26]. During the fabrication process, the soluble PVP is washed out in an aqueous coagulation bath leaving an anisotropic porous PVDF membrane (Fig 1E) [22]. We observed that the resulting PVDF/PVP membranes have a larger pore size and membrane thickness, compared to PVDF-alone membranes (Fig 1F and 1G). We also found that pores formed on the glass exposed side were larger compared to the air exposed side for both

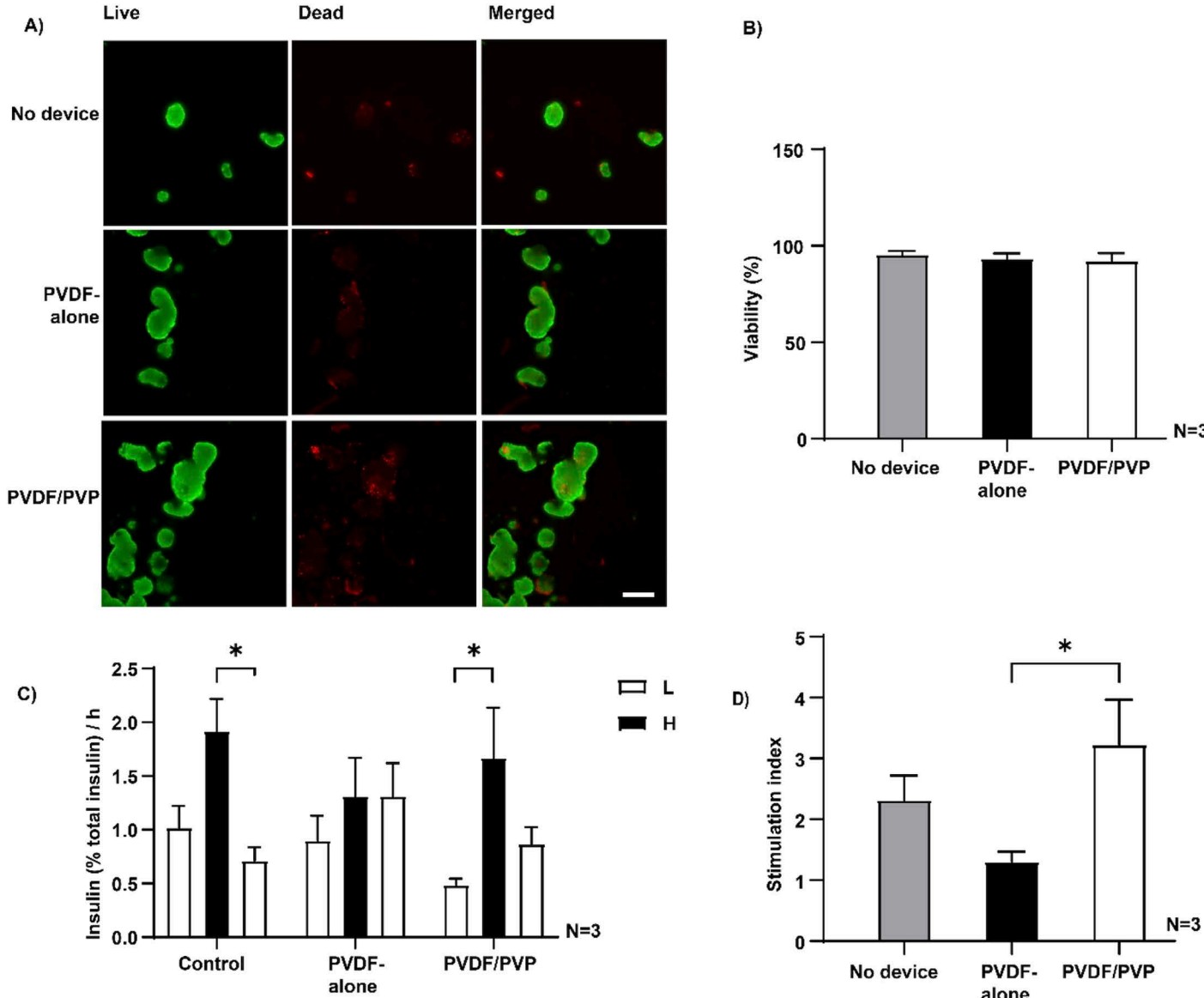

**Fig 7. Human islets cultured inside the macrodevices. (A)** Representative images of human islets inside the encapsulation devices after 8 days of culturing. Red channel represents dead cells, green channel represents live cells. Scale bar represents 200 μm in all pictures. **(B)** Quantification of human islet viability after 8 days of culturing inside the encapsulation devices or without devices as control. Data presented as mean ± SD, n = 3 different donors. **(C)** Insulin release profile of human islets during GSIS after 8 days of culturing inside the devices normalized against total insulin. *P < 0.05, with one way ANOVA. **(D)** Stimulation index at day 8. Data presented as mean ± SEM, n = 3 different donors. *P < 0.05, with Kruskal-Wallis.

membrane types. We speculate that the resulting anisotropic membranes are due to the difference in exposure to water in the coagulation bath, the air side is immediately exposed to water in the coagulation bath, while the glass side is not, leading to a difference in phase separation kinetics on either side as was described by Fontananova *et al* [22]. In simple terms, on the glass side the PVDF/PVP mixture has more time for phase separation, resulting in a different pore geometry containing larger pores on the glass side than on the air exposed site.

Not only membrane pore size and density, but also its hydrophilicity plays a role in the diffusion of molecules of water-soluble molecules such as glucose and insulin. In our study PVDF/PVP membranes were found to be more hydrophilic than PVDF-alone membranes

based on water contact angle measurements (Fig 2A). The decrease in water contact angle after incorporation of PVP was also observed by Mavukkandy *et al.* and Guo *et al* confirming our findings [25,31]. While most of the PVP is washed out from PVDF/PVP membranes, a small fraction remains, seen by the presence of PVP specific bands in FTIR spectra (Fig 3C) [26,31–33]. The increase in hydrophilicity observed in PVDF/PVP membranes is likely caused by these remaining trace amounts of PVP in these membranes.

MTT and LDH assays were performed according to ISO10993-5 medical device standards to examine any cytotoxic effects of these membranes on cells. We chose to use a INS1E insuloma cell line as a model instead of a fibroblast cell line dictated by the ISO protocol to better mimic the device's intended application. Both MTT and LDH levels in INS1E cells cultured in the presence of medium extracts made from PVDF membranes were similar to controls. Moreover, the cell viability was found to be higher than 70%, a threshold generally accepted by international standards for non-cytotoxic biomaterials used in medical devices (Fig 3A and 3B) [34]. Any PVP traces present in PVDF/PVP membranes after manufacturing did not negatively affect cell viability.

For long-term immunoprotection and cell containment of the device, membranes should be mechanically strong and resistant to breakage, thus we measured the mechanical properties of our membranes. We found that the failure stress and strain of break values of the PVDF-alone membrane (Fig 2E and 2F) were in the same order of magnitude as previously reported by Li *et al* [24,35]. The addition of PVP significantly reduced the membrane's Young's modulus and failure stress; however, they remain similar to previously published data of 1.5 MPa [35]. Furthermore, the mechanical properties of skin are greater than those of the PVDF/PVP device, as Ottenio *et al.* reported a skin Young's modulus of 71 MPa [36]. However, as the device will not be a load-bearing implant, we expect that the devices are strong enough for the intended application.

Another device property to consider for implantation is structural integrity. Previously, we have shown that membranes (up to 250 μm thickness) are prone to folding or rolling up which can lead to suboptimal engraftment due to a more severe tissue response [37]. The current PVDF/PVP membranes are approximately 150 μm thick, hence cannot provide sufficient structural support to avoid folding. This issue would be even more noticeable with clinically sized devices. Therefore, we developed a thick ring-like structure made from PVDF, which is incorporated between the two membrane layers without affecting the mechanical strength of the sheets (S4 Fig G). The support ring provides structural support to the device and prevents it from folding while maintaining flexibility. Moreover, the device can now be easily handled with surgical tools (S4 Fig B).

Faster diffusion of nutrients across membranes promotes a fast response to blood glucose and insulin bioavailability. We evaluated the mass transport characteristics of PVDF-alone and PVDF/PVP membranes with fluorescently labelled dextran and insulin, or glucose solutions. The diffusion analysis is based on diffusion of molecules between a high concentration compartment and a no concentration compartment, the driving force of diffusion in this setup is related to the difference between concentrations on either side. The diffusion rate therefore slows down during the experiment until it reaches zero, obviously a different mechanism occurs *in vivo* where insulin or glucose is continuously removed via diffusion into the bloodstream the driving force is expected to be more constant. One has to keep in mind that these *in vitro* experimental conditions might not necessarily reflect insulin diffusion kinetics *in vivo* but, merely serve as an indication how well insulin and glucose and other relevant biomolecules can pass these membranes. In addition, diffusion of insulin from these closed devices is likely helped by the presence of a dense vasculature around and close to the surface of these devices, while the location of implantation might also reflect how well insulin

can diffuse away from these devices. Moreover, the dynamics of diffusion will change over time during the engraftment of the device after its implantation. We observed that diffusion of dextran, glucose, and insulin was improved in PVDF/PVP membranes, compared to PVDF-alone membranes (Fig 4A – I). We observed that insulin diffused slower over the membranes compared to glucose, likely due to the difference in molecular weight, 5.8 kDa of insulin vs 0.18 kDa of glucose [38]. Our finding are similar to the diffusion kinetics reported on the MailPan® device, where 50% of glucose and nearly 40% of insulin had diffused over Polyester membranes with 100 nm pores after 24 h, similar seen in our experiments where an equilibrium for both insulin and glucose was reached after 24 h [39]. Since beta cells rely on oxygen for cell survival and function, we tested whether PVDF-alone and PVDF/PVP membranes allow for relatively unhindered diffusion of oxygen to the encapsulated cells. Since, the diameter of oxygen molecules is around 0.3 nm [40], significantly smaller than the pores of the membranes (<0.4 μm) tested here, oxygen was found to freely diffuse through the PVDF alone and PVDF/PVP membranes (Fig 4H).

To protect the encapsulated beta-cells against cytotoxic antibodies, such as IgG and IgM, or inflammatory cytokines, such as IL-1β, histamine, TNF-α, and TGF-β, the pores should be preferably smaller than 100 nm. However, the downside of a significantly decreased pore size is, that this can hinder diffusion of insulin and glucose [38,41]. Randall *et al.* performed a study where the diffusion of IgG was measured across membranes with different pore diameters ranging from 78 nm to 2 μm. They reported that there was no diffusion of IgG observed at pores of 78 nm, similar to our observations in PVDF-alone membranes with pores around 51 nm (Fig 4L) [42]. We found that PVDF/PVP membranes with 110 nm pores, allow for the diffusion of IgG (Fig 4L and 1H) similar to what is seen in the MailPan® device with a similar pore size, but made from a different biomaterial, indicating that pore size is a significantly limiting factor in closed islet delivery devices [39]. The challenge in engineering macroencapsulation devices for islets and beta cells is to find a balance between an efficient exchange of endocrine hormones, glucose, oxygen and nutrients, physically protecting the encapsulated cells from the host immune cells, and limiting cytokine exposure. The interaction with the recipient's immune cells (cytotoxic T-lymphocytes, macrophages, etc.) is what ultimately destroys the beta cells. Ideally one would like to block both cytokines and immune cells at the same time in macroencapsulation devices. Since beta cells have such a high demand for nutrients and oxygen while a rapid exchange of glucose and insulin is needed to efficiently compensate for metabolic changes related to blood glucose, a compromise has to be found between physical protection and diffusion of relevant molecules. Next, we compared the cell blocking properties of PVDF-alone and PVDF/PVP membranes. We found that macrophages are able to partly migrate through 1 μm pores, but not through 0.45 μm pores (Figure 5), which is in line with previous studies done on other closed devices [29,43]. Loudovaris *et al.* described that in PTFE based macroencapulation devices of a similar design pores below 0.45 μm diameter can be considered immunoprotective, as they prevent both alloimmune and autoimmune destruction of mouse insulinoma cells or islets by blocking immune cells [44]. Based on literature and our findings it's recommended that when engineering macroencapsulation devices for islets and beta cells, one should strive for a balance between maximizing the diffusion of insulin and glucose, while preventing direct cell-cell interactions, and if possible limiting the diffusion of cytotoxic antibodies and inflammatory cytokines as much as possible [4,45].

Finally, we assessed the viability and function of human islets encapsulated inside the devices for up to 8 days. The human islet viability remained high after 8 days of culturing, similar to non-encapsulated islets with a viability of 92 ± 4% (Fig 7A and B). Non-encapsulated islets responded to changing glucose levels after 8 days of culturing with a stimulation index of 2.3

± 0.4 (Fig 7C and D). Human islets encapsulated in PVDF/PVP devices presented a significant higher stimulation index than islet inside PVDF-alone devices, 3.2 ± 0.7 vs 1.3 ± 0.2 (Fig 7C and D). This is the result of the enhanced diffusion capacity of relevant molecules such as insulin and glucose in PVDF/PVP membranes. The stimulation index of the PVDF/PVP device is similar to islets encapsulated for 9 days inside a previous version of the βAir device [46]. Additionally, islets are considered functional if their stimulation index is above 2 [47], we can conclude that the PVDF/PVP device did not diminish the function of the encapsulated islets. Future outlook on clinical translatability of the closed device concept is related to achieving insulin independence while islet isolation centers commonly transplant around 5000 IEQ/kg, this would mean for an 80 kg person, 400.000 IEQ are needed to reach insulin independence. Unfortunately, numerous donor and retrieval factors can influence the outcomes of the islet isolation process and affecting islet yields. Islet isolation centers typically use what they get for transplantation and 5000 IEQ/KG is considered an optimal number but, can sometimes be lower or higher in practice. In light of recent clinical trials going on with devices in the field, multiple smaller devices are probably a more likely strategy used in the future [51]. This ensures additional safety if one of many fails, the others devices can still function. One large device for 400.000 IEQ is not considered a likely strategy in the field.

## Conclusions

In this study, we produced and compared two tailor made macro encapsulation devices made from thin film solvent-non solvent casted membranes from either PVDF, or PVDF in which PVP was added as a supporting pore forming agent. We observed that diffusion of insulin and glucose significantly improved in PVDF/PVP membranes, compared to PVDF-alone membranes, showing that the addition of a porogen during solvent-non solvent film casting improves membrane permeability for these biomolecules. The presence of trace amounts of PVP in PVDF/PVP membranes did not affect cytotoxicity as no significant difference in cell number was observed. Furthermore, both PVDF based membranes can act as a barrier for macrophages, and can be used to block the direct interaction of immune cells with the encapsulated islets on the inside of the devices. Finally, human donor islets in PVDF/PVP devices function better than when encapsulated in PVDF-alone devices likely due to improved glucose and insulin diffusion through PVDF/PVP membranes compared to PVDF-alone membranes. We conclude that a tailor made PVDF/PVP based closed device with its ability to block the entry of immune cells and its enhanced diffusion capacity for relevant biomolecules can act as a potential cell delivery device allogeneic islets and beta cells. The incorporation of a support ring structure at the rim of these devices provides strength and rigidity for improved handling during cell seeding, prolonged cell culture and future implantation studies.

### Mechanical tensile testing support ring

Mechanical properties of the support ring were determined by mechanical tensile testing (Electroforce 3230-ES Series III with 450 kN load cell). Dimensions of test samples were 30 x 5 mm, with an effective area of 5 x 5 mm between the clamps. The ramp speed was 0.1 mm/s. Each condition was tested five times. From stress-strain curves; peak strain, failure stress, failure strain and Young's modulus were calculated.

The strength of the seals, by either heat sealing or ultrasonic welding, were analysed with mechanical tensile testing. The failure stress for both sealing methods were ~ 5 MPa and ~ 0.85 MPa for PVDF-alone and PVDF/PVP membranes respectively (S4 FigG), similar to unsealed membranes (Fig 2E). A decrease in the failure strain was observed in membranes which were welded together with ultrasonic welding, from ~ 70% to ~ 40% for PVDF-alone

membranes and from ~ 11% to ~ 7% for PVDF/PVP membranes (S4 Fig H). The advantage of ultrasonic welding over heat sealing is the flexibility in device design and more control over the welding process, and was therefore used in future implant production.

## Supporting information

**S1 Fig. Effect of changing the casting thickness with respect to final membrane thickness of 15% (w/w) PVDF and 15% (w/w) PVDF + 5% (w/w) PVP membranes.** The red frame represents the selected casting thickness.
(TIFF)

**S2 Fig. Light microscopy images of the bottom of the transwell macrophage migration assay:** After 24h of culturing human primary macrophages in (modified) transwell inserts. Cells were detected in the control and 8 μm wells.
(TIFF)

**S3 Fig. Results of physical properties of PVDF support rings.** (A) Thickness of produced support ring measured from SEM cross-section images. Data presented as mean ± SD, n = 3. (B) SEM cross-section image of PVDF support ring. Scale bar represents 100 μm Mechanical properties of produced membranes (C-F). (C) Peak Stress, (D) Young's modulus, (E) Failure Stress, (F) Failure strain. Data presented as mean ± SD, n = 5. Thickness 208 ± 25 μm, Peak stress 48 ± 4 MPa, Young's' modulus 0.62 ± 0.05 MPa, Failure stress 46 ± 5 MPa, Failure strain 103 ± 10%.
(TIFF)

**S4 Fig. Leakage of encapsulation devices and the mechanical strength of different sealing methods.** (A) Assembled encapsulation device. (B) Handling assembled encapsulation device. Camera shots of device testing for leakage (C-D). (C) Leakage free device and leaking device (D). (E) Heat sealing configuration with two membrane sheets. (F) Ultrasonic welding configuration with one membrane sheet and a support structure. Mechanical properties of heat and ultrasonic welded seals (G-H). (G) Failure stress and (H) failure strain of seals. Data presented as mean ± SD, n = 5. *P < 0.05, with student t-test.
(TIFF)

## Author contributions

**Conceptualization:** Denise F.A. de Bont, Aart Alexander van Apeldoorn.

**Data curation:** Denise F.A. de Bont, Sami G. Mohammed, Omar Paulino da Silva Filho.

**Formal analysis:** Denise F.A. de Bont, Omar Paulino da Silva Filho.

**Funding acquisition:** Eelco J.P. de Koning, Aart Alexander van Apeldoorn.

**Investigation:** Denise F.A. de Bont, Sami G. Mohammed, Vijayaganapathy Vaithilingam, Marlon J. Jetten.

**Methodology:** Denise F.A. de Bont, Sami G. Mohammed, Rick H.W. de Vries, Omar Paulino da Silva Filho, Vijayaganapathy Vaithilingam, Aart Alexander van Apeldoorn.

**Project administration:** Denise F.A. de Bont, Aart Alexander van Apeldoorn.

**Resources:** Marten A. Engelse, Eelco J.P. de Koning.

**Supervision:** Aart Alexander van Apeldoorn.

**Visualization:** Denise F.A. de Bont, Sami G. Mohammed.

**Writing – original draft:** Denise F.A. de Bont.

**Writing – review & editing:** Denise F.A. de Bont, Sami G. Mohammed, Rick H.W. de Vries, Omar Paulino da Silva Filho, Vijayaganapathy Vaithilingam, Marlon J. Jetten, Marten A. Engelse, Eelco J.P. de Koning, Aart Alexander van Apeldoorn.

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
