## [Decision Letter · Decision Letter 0]

10 Oct 2022

PONE-D-22-17075Supporting islet function in a PVDF membrane based immunoprotective cell delivery device by solvent non-solvent casting using PVPPLOS ONE

Dear Dr. van Apeldoorn,

Thank you for submitting your manuscript to PLOS ONE. After careful consideration, we feel that it has merit but does not fully meet PLOS ONE’s publication criteria as it currently stands. Therefore, we invite you to submit a revised version of the manuscript that addresses the points raised during the review process.

We look forward to receiving your revised manuscript.

Kind regards,

Sabata Martino, Ph.D

Academic Editor

PLOS ONE

Journal Requirements:

2. Please provide the following information regarding tissue/organ donors analyzed in your study.

a. Please provide the source(s) of the transplanted tissue/organs used in the study, including the institution name and a non-identifying description of the donor(s).

b. Please state in your response letter and ethics statement whether the transplant cases for this study involved any vulnerable populations; for example, tissue/organs from prisoners, subjects with reduced mental capacity due to illness or age, or minors.

- If a vulnerable population was used, please describe the population, justify the decision to use tissue/organ donations from this group, and clearly describe what measures were taken in the informed consent procedure to assure protection of the vulnerable group and avoid coercion. 

- If a vulnerable population was not used, please state in your ethics statement, “None of the transplant donors was from a vulnerable population and all donors or next of kin provided written informed consent that was freely given.”

c. In the Methods, please provide detailed information about the procedure by which informed consent was obtained from organ/tissue donors or their next of kin. In addition, please provide a blank example of the form used to obtain consent from donors, and an English translation if the original is in a different language.

d. Please indicate whether the donors were previously registered as organ donors. If tissues/organs were obtained from deceased donors or cadavers, please provide details as to the donors’ cause(s) of death.

e. Please discuss whether medical costs were covered or other cash payments were provided to the family of the donor. If so, please specify the value of this support (in local currency and equivalent to U.S. dollars).

Yes

This work was funded by the partners of Regenerative Medicine Crossing Borders (RegMed XB), a public-private partnership that uses regenerative medicine strategies to cure common chronic diseases. This collaboration project is financed by the Dutch Ministry of Economic Affairs by means of the PPP Allowance made available by the Top Sector Life Sciences & Health to stimulate public-private partnerships.

https://regmedxb.com/

Aart van Apeldoorn and Denise de Bont are coauthors on a patent currently in review on an “Immunoprotective type implantable cell delivery device” submitted by Maastricht University

Reviewers' comments:

Reviewer's Responses to Questions

**Comments to the Author**

1. Is the manuscript technically sound, and do the data support the conclusions?

Reviewer #1: Partly

Reviewer #2: Partly

Reviewer #3: Partly

Reviewer #4: Partly

2. Has the statistical analysis been performed appropriately and rigorously? 

Reviewer #1: Yes

Reviewer #2: No

Reviewer #3: No

Reviewer #4: Yes

3. Have the authors made all data underlying the findings in their manuscript fully available?

Reviewer #1: No

Reviewer #2: Yes

Reviewer #3: Yes

Reviewer #4: Yes

4. Is the manuscript presented in an intelligible fashion and written in standard English?

Reviewer #1: Yes

Reviewer #2: Yes

Reviewer #3: Yes

Reviewer #4: Yes

5. Review Comments to the Author

Reviewer #1: In their manuscript ‘supporting islet function in a PVDF membrane based immunoprotective cell delivery device by solvent non-solvent casting using PVP’ the authors report on the construction of a PVDF microencapsulation device for islet cell transplantation. They enhanced permeability of the devices thanks to the incorporation of PVP while casting the membrane.

The manuscript is well-written and well prepared. The devices are well characterized by their mechanical and biological properties. Functional data are however less clear.

Moreover, a number of devices constructed with different biomaterials are under development or used for preclinical and clinical testing with islet cell tissue of different origin. It is unclear for the reader of the manuscript how this device can be used as a possible alternative while also focusing on the difficulties that are encountered during in vivo application of different devices for macro-encapsulation.

Some points may be addressed to clarify this:

The authors chose using PVDF/PVP as biomaterial for islet cell encapsulation because of lower oxidative stress to alpha and beta cells; however, strong data on biocompatibility, especially for the purpose of cell encapsulation are missing. Do the authors have data on the degree of pericapsular fibrotic overgrowth when using this type of device? It would be valuable to add this to the manuscript.

How does the PVDF/PVP biomaterial differs from PTFE, which is used in most of the clinical applied devices in terms of mechanical, functional and biocompatible properties? Comparisons of diffusion properties and biomaterials with clinical used materials would be of added value to the manuscript.

Islet cells need to respond quickly to increase in blood glucose levels; this might be a disadvantage of the devices since diffusion of glucose (equilibrium after 3h)and insulin (equilibrium after 24 hours) seems rather slow. Can the authors comment on this?

It is unclear how this slow diffusion affects the real glucose stimulation index: the comparison between the free islet cells with the encapsulated ones is difficult to make since number of IEQ and media concentrations seem different; it would be interesting to know basal and stimulated insulin synthesis per IEQ or beta cell. Were experiments of different conditions performed in parallel with tissue of the same donor? How was the number of IEQ determined?

Adding perifusion experiments with devices and free cells would be of interest.

Macro-encapsulation devices with islet cells from different sources (allo, xeno, iso, stem cell generated) have been shown to restore metabolic function in immune deficient rodent models; this is necessary to confirm possible in vivo islet cell function within the devices. Are in vivo data available?

Some minor remarks and questions:

The authors use islets at 75% purity; how was this determined?

The authors examine the immune-protective capacity of the membranes by evaluating the degree of macrophage migration; this is however only a portion of immune-reactivity. Could the authors comment on this?

How was the pore diameter calculated?

In the introduction the authors refer to one of their previous studies using directly vascularized polymer-based cell delivery devices transplanted in the fat pad; however, the correct reference is missing in the manuscript.

How many cells could be loaded maximally in a device without causing cell death?

One restriction of the use of macrodevices is the number of cell material that can be loaded inside the device. How many islet cells can be injected in the devices proposed in the manuscript? At what cell density could the cells survive and function in vivo?

Reviewer #2: Dear author,

Thank you for submitting your manuscript “Supporting islet function in a PVDF membrane based immunoprotective cell delivery device by solvent non-solvent casting using PVP”.

I read with interest your paper and I believe that with some additional work it would be a valuable contribution to the literature.

However, I believe that in the current form, it is not suitable for publication.

The study presents the results of original research but experiments, statistics, and other analyses are not performed to a high technical standard, not described in sufficient detail and conclusions are not fully supported by the data presented.

While the mechanical characterization of the materials is sufficiently described, it is not clear the implication of having different pore sizes on different surfaces of the membranes. How does this affect the device assembly? Moreover, a lot of importance is given to the fact that the PVDF/PVP membranes allowed a greater diffusion of dextran, glucose, insulin and IgG compared to PVDF alone. Nevertheless, the fact that IgG diffuses through the membrane is an aspect that deserves more investigation. Even though the immunoprotection ensured by the size of the pores prevents the migration of macrophages (which is already known in the literature, as suggested by the author), it is not discussed whether this IgG diffusion could be detrimental to the islets' viability. Additionally, also “the permeability to inflammatory cytokines, such as IL-1β, histamine, TNF-α, and TGF-β” should be taken into consideration. Experiments in which these molecules are used during the culture of encapsulated islets and their effect on viability and function are likely to be included.

Similarly, co-culture systems with islets inside the device and macrophages outside could be helpful to confirm that they do not harm the islets.

In summary, I believe that further experiments on the biological validation of using this technology are necessary to complete this paper, ideally supported with results on their in vivo application.

I encourage the authors to resubmit the manuscript after performing new experiments to demonstrate the scientific validity of their technology.

Reviewer #3: The manuscript titled: ‘’Supporting islet function in a PVDF membrane based immunoprotective cell delivery device by solvent non-solvent casting using PVP’’ by Denise F.A. de Bont et al., focuses on the optimization of membrane fabrication PVDF-PVP ratio with respect to the diffusion properties of the device that can be tailored to support human islet function while preventing at the same time the membrane entry of primary human macrophages that damaged beta cells.

Even if the study presented the findings of original research focusing on new therapeutic approaches for Type 1 diabetic (T1D) patients who are lifelong dependent on insulin therapy to keep their blood glucose levels under control, the experiments, particularly the in vitro experiment, and other analyses were not performed to a high technical standard and were not presented and described in sufficient detail. Consequently, the conclusions are not supported by the appropriate data. A large quantity of data must be added to support the authors' hypothesis, which now is missing from the text.

Major points:

-There is a need of a test that validates the presence of pure islets after culture on the device (e.g., the Dithizone assay), as well as an evaluation of their functioning throughout time in culture and morphologic characterization. I suggest the gene expression characterization for the islets that are cultured in PVDF-PVP device at short and long time in culture.

-In the manuscript are used INS-1E rat insulinoma cells, machophages and human islets. While the function of using macrophages is obvious, it is not clear why, in the last paragraph, human islet cells are used in place of the INS-1E rat insulinoma cells that were previously studied for cytotoxicity of membranes. In the paragraph and the discussion section, I suggest a more thorough explanation of the experimental strategy.

-The goal of the manuscript is preventing the entry of primary human macrophages that damage beta cells with the use of this device, but in the paragraph ‘’diffusion of small molecules through membranes’’ the authors demonstrated the capacity of IgG to pass through the membrane. This event could damage the efficacy of the human islet functionality? I suggest discussing this point and include new experiments that verified this point.

-Moreover, I suggest reviewing the material and method section. Experiments are not described in an appropriate manner. in particular in the MTT assay paragraph, what do you mean for extraction samples? Are they the device with the different cells or the cells are removed from the device and then plated alone on the 96 well in order to obtain 50% of the confluent monolayer?

Consider explaining the experiments more carefully.

Minor points:

-In the manuscript are present some mistakes (e.g. line 172). I request a spell-check of the text.

Reviewer #4: Paper entitle “Supporting islet function in a PVDF membrane based immunoprotective cell delivery device by solvent non-solvent casting using PVP” PONE-D-22-17075 appears to be carefully and scientifically conducted with regard to the fabrication and characterization of the proposed new material the PVDF/PVP. Unfortunately, the whole part concerning the interaction of the device with islets or cells for the purpose to demonstrating its safety and immunoprotective capabilities does not appear to be appropriate either as a methodology or as an interpretation of the results. The data presented are in no way sufficient to define the device as immunoprotective. The only data Authors show is that passively macrophages (moreover, cultured under conditions that do not adhere to the vendor's recommendations) do not pass through the pores created on the PVDF/PVP. Confusion is also made between human islets and beta cells: it is one thing to use whole islets taken from cadavers and another to use beta cells purified from them. The work should be thoroughly reviewed and presented as a characterization of the material and first circumstantial in vitro evaluation of the safety of the device. To demonstrate the immunoprotective capabilities of a new material or device many other evaluations must be added (e.g., in vitro co-culture device empty or with islets and PBMCs from donors, followed by lymphocyte typing with cytokine and chemokine assay, macrophage migration assays with device containing chemoattractant first and then human islets, transplantation into animal models first immunocompromised for assessment of safety and foreign body reaction in vivo and then immunocompetent of the empty device and then containing islets, lymphomonocyte caracterization of transplanted animals, cytokine and chemokine assays under the various conditions, etc... ).

The discussion is by no means exhaustive, some concepts are already said in the introduction and therefore do not need to be recalled in this section. In several places it appears confusing because it is difficult to understand whether the authors are referring to their own data or to the literature, and often comparisons with devices presented by other groups do not seem appropriate. Also, the issue of where to transplant the device and the number/surface of devices needed for the correction of diabetic hyperglycemia are not addressed, it would be useful to address this issue to fully understand the potential of the proposed device.

The various critical issues are detailed below:

Major revisions

Title “Supporting islet function in a 1 PVDF membrane based immunoprotective cell delivery device by solvent non-solvent casting using PVP” It is not appropriate to talk about immunoprotection based on the data provided; the title should be changed.

Lines 39-41 “We report on the creation of porous polyvinylidene fluoride (PVDF) membrane-based devices for islet and beta-cell transplantation.” In the paper, the authors do not test beta cell transplantation in any way; these should be removed from this and other sentences.

Lines 45-47 “We showed that the use of a water-soluble pore forming reagent, PVP, can significantly increase glucose diffusion through these membranes without compromising their immunoprotective properties.” The immunoprotective properties of the device are not demonstrated by the data presented and therefore the sentence should be changed.

Lines 55-56 “….the membrane can still serve as an immunoprotective barrier preventing the entry of primary human macrophages and damaging beta cells Immunoprotection is quite a different concept and must be defined by very specific methods . Macrophages may remain outside but there is nothing to prevent them from being activated and secreting cytokines and chemokines .

Lines 96-97 “Here we report on the development of an immunoprotective extrahepatic islet delivery device....” It is not correct to talk about “extrahepatic” sites instead alternative implantation sites on which to graft the device should be clearly defined.

Line 156 Explain the rationale for the use of INS-1E cells. If they are used as a control of whole insulae, they are not very comparable because they represent tumor cells derived from beta cells only.

Line 232- “The donor chamber was filled with a PBS solution containing 2 mg/mL 3-5 kDa FITC-dextran (Sigma-Aldrich, the Netherlands), 20 mM D-glucose solution …… “ The method described is not clear it seems that everything is put together at the same time in the same donor chamber. Rewrite the sentence.

Line 266 and figures 4K) 4L). “….that is free-floating control…” What is meant by this control? In all other analyses, PVDF alone was compared with PVDF/PVP. Please explain further and correct.

Line 269 “Immunoprotective capacity of membranes” The title of the paragraph is inappropriate. From what is reported, the test is carried out using a device that does not contain any kind of biological material, so what is being tested is solely the ability of macrophages to react to the presence of the membrane. A more effective motility and invasiveness test would require the presence in the device of cells or molecules that can specifically attract macrophages toward a chemo-attractant gradient.

In addition The 10% serum itself is used as a chemo-attractant for macrophages. So why are macrophages not cultured in serum free medium for this test?

Furthermore, in the paragraph on culturing macrophages (lines165-166) it is stated that they are cultured in "in Dulbecco's Modified Eagle Medium (DMEM) high glucose supplemented with 10% (v/v) FBS"....but the specifications of the company from which they are purchased recommend using pre-coated flasks with “Human Macrophage Primary Cell Culture Complete Extra-cellular Matrix”. Please explain the reasons for this type of culture.

Line 285 Why do the authors use different islets numbers between the device and the control ? When comparing results, even calculating them as a stimulation index could lead to errors. It would be a good idea to use the same IEQ number or explain the reasons for the choice made.

Line 294 Viability should also be done at time 0, when the device is refilled, for a more appropriate comparison At the end of the experiment how many islets are recovered of the 3,000 IEQs originally put in ?

Line 379 Cytotoxicity should also be assessed on whole islets not only on INS.

Line 382 Figure 3A not 4A

Line 385 Figure 3B not 4B

Line 418 “FITC-labelled insulin reached equilibrium after 24 h in PVDF/PVP membranes as 47 ± 2% of insulin was diffused, compared to whereas 28 ± 5% was diffused in PVDF-alone membranes at the same time point (Fig. 4G).” Twenty-four hours is a really long time to reach equilibrium in PVDF/PVP membranes especially if you want to consider implanting this device in vivo. The islets inserted in the device are able to sense in real time the glucose present in the culture medium but the insulin they release is not able to diffuse from the device as fast, this could compromise metabolic control. The graph of static incubation presented as an index is not sufficient to understand whether the proportion of insulin produced is proportional to the glucose administered, it only shows that islets respond to the increase in glucose concentration even in the device. It should be discussed appropriately in discussion.

Line 446 “ Immunoprotective function of PVDF membranes” Title is inappropriate, with the illustrated experiment only the ability of macrophages to spontaneously pass through holes is determined but not the immunoprotective character of the device.

Line 503 “Immunoprotective beta-cell delivery devices…” the Device is tested only on whole islets from donor cadaver there is no data on purified beta-cells correct or enter corresponding data. The word “Immunoprotective” is not used correctly and should be changed.

“Discussion” : Some concepts are already said in the introduction and therefore need not be recalled in this section (line 517 for example). It needs to be rewritten considering previous recommendations. Also, the possible implantation site should be discussed prospectively, in number of devices to be used based on the number of IEQs it contains. The device presented is an oval of 44X28 mm, containing 3000 IEQs. To achieve blood glucose correction in an 80 kg diabetic man, it is estimated that at least 1000000 IEQs are needed...so could the device's size be substantial?

Line 597 and Line 599 figure cited (Fig. 4L) refers to the area under the oxygen diffusion curve.

Lines 598-599 “On the other hand, PVDF/PVP membranes with 110 nm pores, similar to the MailPan® devices with 100 nm pores, allow the diffusion of IgG (Fig. 4L and 1H).39 Next, we compared the immune protection” Is there a contradiction with what was said above? Line 425: “For IgG only 1.6 ± 0.7 % permeated through the PVDF/PVP membrane, but IgG could not be detected in the receiver chamber of the PVDF-alone membrane, suggesting no diffusion (Fig.4J)”

Line 604 Authors do not demonstrate the device's ability to block the circulation of pro-inflammatory chemokines and cytokines.

Minor revisions:

Line 89-91 “Based on these findings, we selected polyvinylidene fluoride (PVDF) since it induces relatively low oxidative stress in beta-cells compared to the other tested biomaterial….” Add references.

Lines 106-107 “We showed that primary human islets remained viable and functional for 8 days, like free-floating controls.” Have other times been evaluated ?

Line 320 and Line 481 and FIG7C;7D In addition to the stimulation index, values of insulin concentration per microgram of total DNA or protein should also be reported.

Page 19 Figure 1 G,H,I It is preferable to use the same units on the ordinates for clarity.

Page 24 Figures 4 B) 4 E) check the scale in the ordinate since it does not start from zero as with the other graphs.

6. PLOS authors have the option to publish the peer review history of their article (what does this mean? ). If published, this will include your full peer review and any attached files.

**Do you want your identity to be public for this peer review?** For information about this choice, including consent withdrawal, please see our Privacy Policy .

Reviewer #1: No

Reviewer #2: No

Reviewer #3: No

Reviewer #4: No

---

## [Author Response · Author response to Decision Letter 0]

11 Dec 2023

Dear editorial office,

We have read the comments of all reviewers and made adjustments accordingly. We have also adressed all the additions/alterations requested by your office

The changes based on the reviewers’ comments and suggestions are addressed one by one and a description underneath every question.

I have indicated textual changes by highlighting new text in the manuscript in yellow to help discern these issues.

All the best,

Aart van Apeldoorn

Assistant professor

MERLN institute for technology-inspired regenerative medicine

Maastricht University

The Netherlands

Editorial office comments:

Please state what role the funders took in the study. If the funders had no role, please state:

We added this statment to the funders section

Aart van Apeldoorn and Denise de Bont are coauthors on a patent currently in review on an “Immunoprotective type implantable cell delivery device” submitted by Maastricht University

Both issues have been adressed and mentioned in the corresponding section

3. Please provide the following information regarding tissue/organ donors analyzed in your study.

a. Please provide the source(s) of the transplanted tissue/organs used in the study, including the institution name and a non-identifying description of the donor(s).

b. Please state in your response letter and ethics statement whether the transplant cases for this study involved any vulnerable populations; for example, tissue/organs from prisoners, subjects with reduced mental capacity due to illness or age, or minors.

- If a vulnerable population was used, please describe the population, justify the decision to use tissue/organ donations from this group, and clearly describe what measures were taken in the informed consent procedure to assure protection of the vulnerable group and avoid coercion.

- If a vulnerable population was not used, please state in your ethics statement, “None of the transplant donors was from a vulnerable population and all donors or next of kin provided written informed consent that was freely given.”

c. In the Methods, please provide detailed information about the procedure by which informed consent was obtained from organ/tissue donors or their next of kin. In addition, please provide a blank example of the form used to obtain consent from donors, and an English translation if the original is in a different language.

d. Please indicate whether the donors were previously registered as organ donors. If tissues/organs were obtained from deceased donors or cadavers, please provide details as to the donors’ cause(s) of death.

e. Please discuss whether medical costs were covered or other cash payments were provided to the family of the donor. If so, please specify the value of this support (in local currency and equivalent to U.S. dollars).

These above issues have been adressed an are now elborately described in the materials and methods section.

Reviewer #1: In their manuscript ‘supporting islet function in a PVDF membrane based immunoprotective cell delivery device by solvent non-solvent casting using PVP’ the authors report on the construction of a PVDF microencapsulation device for islet cell transplantation. They enhanced permeability of the devices thanks to the incorporation of PVP while casting the membrane.

The manuscript is well-written and well prepared. The devices are well characterized by their mechanical and biological properties. Functional data are however less clear.

Moreover, a number of devices constructed with different biomaterials are under development or used for preclinical and clinical testing with islet cell tissue of different origin.It is unclear for the reader of the manuscript how this device can be used as a possible alternative while also focusing on the difficulties that are encountered during in vivo application of different devices for macro-encapsulation.

We thank the reviewer for the time and comments. We improved on the clarification and motivation and discussion about macro devices throughout the entire manuscript.

Some points may be addressed to clarify this:

The authors chose using PVDF/PVP as biomaterial for islet cell encapsulation because of lower oxidative stress to alpha and beta cells; however, strong data on biocompatibility, especially for the purpose of cell encapsulation are missing. Do the authors have data on the degree of pericapsular fibrotic overgrowth when using this type of device? It would be valuable to add this to the manuscript.

We highly appreciate the reviewer’s suggestion and we rewrote some parts of the introduction to clarify this better and have a clearer motive for the study for the readers to understand. We have done an extensive biocompatibility study done in small animals after this manuscript was written and submitted to PLOS. We choose not to include this study since placing both studies into one paper would also change the entire scope of this paper and prevent us from discussion both studies in depth. This manuscript focused on the design and fine tuning of membranes used for device manufacturing prior to extensive animal studies

How does the PVDF/PVP biomaterial differs from PTFE, which is used in most of the clinical applied devices in terms of mechanical, functional and biocompatible properties? Comparisons of diffusion properties and biomaterials with clinical used materials would be of added value to the manuscript.

We thank the reviewer for this question and we understand where this is coming from, since there is commercially device available for research (Theracyte�) composed of PTFE. It is hard to make a direct comparison between these biomaterials simply because they are different in nature and need to be processed in a completely different manner to create membranes from. Any comparison would likely show they are different but that is by definition the case. However, we addressed some of these differences in the introduction and added additional plus references to highlight this a bit better.

Islet cells need to respond quickly to increase in blood glucose levels; this might be a disadvantage of the devices since diffusion of glucose (equilibrium after 3h)and insulin (equilibrium after 24 hours) seems rather slow. Can the authors comment on this?

Very relevant comment and a topic continuously discussed in the T1D field. We have written an additional explanation is added to the discussion.

The diffusion setup is based on diffusion of molecules between a high concentration compartment and no concentration compartment the driving force of diffusion in this setup is related to the difference between concentrations on either side. The diffusion rate therefore slows down during the experiment until it reaches zero, one can imagine a different mechanism occurring in vivo where the insulin or glucose is continuously removed the driving force is expected to stay high.

It is unclear how this slow diffusion affects the real glucose stimulation index: the comparison between the free islet cells with the encapsulated ones is difficult to make since number of IEQ and media concentrations seem different; it would be interesting to know basal and stimulated insulin synthesis per IEQ or beta cell. Were experiments of different conditions performed in parallel with tissue of the same donor? How was the number of IEQ determined?

We thank the reviewer for the comments, these are multiple questions we address here one by one. We chose to measure insulin secretion compared to total insulin present in all beta cells in each condition. One can assume that beta cells contain more or less a similar amount of insulin granules. This relative measurement compensates for any differences between number of IEQs used per experiment like one would use DNA or IEQ.

One islet equivalent (IEQ) is an islet of diameter 150 μm, typically containing 1,500-2,000 cells, including 40-60% beta cells. Islets of different sizes are volumetrically adjusted to IEQs for each experiment

Yes, experiments of the different conditions where done in parallel with donor tissue derived from the same donor during each experiment, otherwise donor to donor variation would overshadow any of the outcomes. In the end outcomes with the different donors where averaged and depicted in the relevant graphs as such. We have added some additional explanation in the caption of the graphs to clarify this

Adding perifusion experiments with devices and free cells would be of interest.

We agree with the reviewer that this could be an interesting addition. However, afterwards repeating this would mean new human donor islets would have to be required, a minimal of 3 is needed while human donor islets are scarcely available, and all assays for all the other in vitro analyses would also have to be repeated. We have used GSIS comparable to others publications in the field to show that insulin comes out in similar quantities as we see for free floating non encapsulated cells. This is why a standard GSIS has been used similar to previous publications on this topic. A perfusion assay could indeed be used to do a in line time-based study, however one has to realize that these dynamics likely completely change in vivo.

Macro-encapsulation devices with islet cells from different sources (allo, xeno, iso, stem cell generated) have been shown to restore metabolic function in immune deficient rodent models; this is necessary to confirm possible in vivo islet cell function within the devices.

Are in vivo data available?

Unfortunately, not yet, since this paper is about a first concept, a new design using a particular membrane fabrication technique to create porous membranes from which a macroencapsulation device can be made. We focused thus primarily on the characterization and fabrication of the device rather than in vivo studies to make sure we understand the working principle prior to elaborate in vivo studies . We have some preliminary data with human islets in a non-diabetic mouse, showing the presence of human c-peptide. However, the number of animals used does not provide enough power for proper statistical analysis and can only be considered as anecdotal observations. A more elaborate study based on this paper would be a next step. We have been working towards a long-term study in a rat model. However, these studies are still ongoing and also would change the focus of this manuscript.

Some minor remarks and questions:

The authors use islets at 75% purity; how was this determined?

As described in the material and methods, human donor islets were provided by our colleagues from the Leiden University medical center’s clinical islet isolation and transplantation center. Islet purity is commonly determined in the field by a standard DTZ staining and microscopic evaluation. DTZ stains zinc associated to insulin red, which can be used to distinguish islets from non-endocrine tissue in isolation preps using light microscopy. This is a commonly used method to determine purity of islet preps all over the world in research and clinical applications. We didn’t feel therefor that it’s necessary to describe the determination of purity elaborately in detail since this is standard practice for releasing islets. We’ve added an additional sentence to the M&M to clarify this further.

The authors examine the immune-protective capacity of the membranes by evaluating the degree of macrophage migration; this is however only a portion of immune-reactivity. Could the authors comment on this?

We have added a revised paragraph on cytokines and IgG in which we have changed the wording related to blocking the entry of macrophages into the device to clarify more about this topic.

How was the pore diameter calculated?

We apologize for not describing this clearly in the M&M. We have added a description in the M&M to clarify this. Pore diameter was determined by SEM measurements on the surface of the membranes. For example, in figures 1 and 4 the pores can be clearly seen on the surface of all samples used. One can use the measurement tool in the software of the SEM to determine the dimensions

In the introduction the authors refer to one of their previous studies using directly vascularized polymer-based cell delivery devices transplanted in the fat pad; however, the correct reference is missing in the manuscript.

We have inserted a reference for this omission

How many cells could be loaded maximally in a device without causing cell death? One restriction of the use of macrodevices is the number of cell material that can be loaded inside the device. How many islet cells can be injected in the devices proposed in the manuscript? At what cell density could the cells survive and function in vivo?

Interesting questions indeed. However at this moment we can not say for sure. We agree with the reviewer this could be an interesting study and should be done in general for all macrodevices. This would require an extensive study in oxygen distribution versus islet function and cell survival. If one wants to do a complete study, it should include probably beta cells in suspension, pseudoislets, stem cell derived islets and primary islets from multiple human donors and perhaps even different animal species to provide a precise answer. In addition, such a study should maybe also include a head-to-head comparison with other macrodevices. The maximum volume which can be injected into these rat sized devices is about 1.5 ml. Based on our own modelling studies and of others islets with a diameter of 150 micrometer are optimal in terms of hypoxia and the distance between them can still be quite small ~40 micrometers before detrimental oxygen levels are reached. However, it can make a huge difference if one seeds islets or single cells in macrodevices the cell density might be way higher with individual cells, but then competition for the necessary nutrients can induce a significant amount of cell death. Stem cell derived beta cells generally cope better with suboptimal oxygen levels due to their immaturity compared to primary beta cells. Also the location of implantation might have a strong effect on cell survival, lack of vascularization exacerbates the lack of oxygen. If needed we can add this discussion to the discussion paragraph provided it doesn’t make the manuscript too long.

Reviewer #2: Dear author,

Thank you for submitting your manuscript “Supporting islet function in a PVDF membrane based immunoprotective cell delivery device by solvent non-solvent casting using PVP”.

I read with interest your paper and I believe that with some additional work it would be a valuable contribution to the literature.

However, I believe that in the current form, it is not suitable for publication.

The study presents the results of original research but experiments, statistics, and other analyses are not performed to a high technical standard, not described in sufficient detail and conclusions are not fully supported by the data presented.

While the mechanical characterization of the materials is sufficiently described, it is not clear the implication of having different pore sizes on different surfaces of the membranes. How does this affect the device assembly?

Interesting question, but it doesn’t, the pore sizes are in the submicrometer range and can hardly be seen by the naked

---

## [Editor Report · Decision Letter 1]

20 Jan 2024

Supporting islet function in a PVDF membrane based immunoprotective cell delivery device by solvent non-solvent casting using PVP

PONE-D-22-17075R1

Dear Dr. van Apeldoorn,

We’re pleased to inform you that your manuscript has been judged scientifically suitable for publication and will be formally accepted for publication once it meets all outstanding technical requirements.

Kind regards,

Sabata Martino, Ph.D

Academic Editor

PLOS ONE

Additional Editor Comments (optional):

no comments
---

## [Editor Report · Acceptance letter]

PONE-D-22-17075R1

PLOS ONE

Dear Dr. van Apeldoorn,

I'm pleased to inform you that your manuscript has been deemed suitable for publication in PLOS ONE. Congratulations! Your manuscript is now being handed over to our production team.

Kind regards,

on behalf of

Prof. Sabata Martino

Academic Editor

PLOS ONE